EMBO
Molecular Medicine

# Upfront admixing antibodies and EGFR inhibitors preempts sequential treatments in lung cancer models

Ilaria Marrocco[1], Donatella Romaniello[1†], Itay Vaknin[1], Diana Drago-Garcia[1] (iD), Roni Oren[2],
Mary Luz Uribe[1] (iD), Nishanth Belugali Nataraj[1] (iD), Soma Ghosh[1‡], Raya Eilam[2], Tomer-Meir Salame[3],
Moshit Lindzen[1] & Yosef Yarden[1,*] (iD)

## Abstract

**Some antibacterial therapies entail sequential treatments with different antibiotics, but whether this approach is optimal for anticancer tyrosine kinase inhibitors (TKIs) remains open. EGFR mutations identify lung cancer patients who can derive benefit from TKIs, but most patients develop resistance to the first-, second-, and third-generation drugs. To explore alternatives to such whack-a-mole strategies, we simulated in patient-derived xenograft models the situation of patients receiving first-line TKIs. Monotherapies comprising approved first-line TKIs were compared to combinations with antibodies specific to EGFR and HER2. We observed uniform and strong superiority of all drug combinations over the respective monotherapies. Prolonged treatments, high TKI dose, and specificity were essential for drug–drug cooperation. Blocking pathways essential for mitosis (e.g., FOXM1), along with downregulation of resistance-conferring receptors (e.g., AXL), might underlie drug cooperation. Thus, upfront treatments using combinations of TKIs and antibodies can prevent emergence of resistance and hence might replace the widely applied sequential treatments utilizing next-generation TKIs.**

**Keywords** EGFR TKIs; first-line therapy; mAbs; NSCLC; resistance
**Subject Categories** Cancer; Respiratory System

## Introduction

Lung cancer is one of the most common malignancies and the leading cause of oncology-related death worldwide. More than 75% of all lung cancer cases are classified as non-small cell lung cancer

(NSCLC) (Sun *et al*, 2007). A variable fraction (12-30%) of NSCLC patients presents activating mutations in the epidermal growth factor receptor gene (*EGFR*) (Lynch *et al*, 2004; Paez *et al*, 2004; Pao *et al*, 2004; Wang *et al*, 2011; Zhang *et al*, 2016). The presence of these mutations confers sensitivity to EGFR-specific tyrosine kinase inhibitors (TKIs). Erlotinib and gefitinib, the first-generation TKIs, achieved clear superiority, in comparison with chemotherapy, in terms of progression-free survival (PFS) (Mitsudomi *et al*, 2010; Zhou *et al*, 2011; Rosell *et al*, 2012). Despite initial good responses to the first-generation TKIs, patients inevitably become resistant within 10–14 months. The most common mechanisms of resistance involve a secondary EGFR mutation (T790M) (Pao *et al*, 2005), overexpression of AXL (Zhang *et al*, 2012), and amplification of *MET* (Engelman *et al*, 2007) or *HER2* (Takezawa *et al*, 2012).

Second-generation EGFR-specific TKIs, afatinib and dacomitinib, irreversibly bind with the ATP-binding cleft of EGFR. A third-generation irreversible inhibitor, osimertinib, targets not only *EGFR* exon 19 deletions and L858R, but also the major resistance mutation, T790M. The results of the AURA3 trial, which compared osimertinib with chemotherapy in T790M-positive patients, led to the approval, in 2015, of osimertinib as a second-line treatment (Mok *et al*, 2017). Three years later, osimertinib was approved as a first-line treatment based on improved PFS compared with patients who received gefitinib or erlotinib (Soria *et al*, 2018). Recent data showed that the most common mechanisms of resistance to osimertinib in first-line settings are *MET* amplification, the C797S mutation, *HER2* amplification, and mutations in downstream signaling proteins (Ramalingam *et al*, 2018). Thus, despite the availability of five EGFR-specific TKIs, the long-term efficacy of these drugs is limited by inevitable emergence of resistance.

In cancer therapy, monoclonal antibodies (mAbs) are largely used in combination with other treatments (Marrocco *et al*, 2019). Hence, combinations of TKIs and mAbs might offer strategies to overcome recurring resistance. For example, a fourth-generation

---

1 Department of Biological Regulation, Weizmann Institute of Science, Rehovot, Israel
2 Department of Veterinary Resources, Weizmann Institute of Science, Rehovot, Israel
3 Department of Life Sciences Core Facility, Weizmann Institute of Science, Rehovot, Israel
 *Corresponding author. Tel: +972 8 934 3974; Fax: +972 8 934 2488; E-mail: yosef.yarden@weizmann.ac.il
 †Present address: Department of Experimental, Diagnostic and Specialty Medicine-DIMES, Alma Mater Studiorum University of Bologna, Bologna, Italy
 ‡Present address: Department of Thoracic Head and Neck Medical Oncology, Division of Cancer Medicine, MD Anderson Cancer Center, Houston, TX, USA

EGFR inhibitor showed synergy in mouse models when combined with cetuximab, a clinically approved anti-EGFR mAb (Jia *et al*, 2016). A combination of erlotinib and cetuximab, which was tested in a phase I/II clinical trial in erlotinib-resistant patients, showed no improvement in terms of PFS (Janjigian *et al*, 2011), and, likewise, disappointing results were observed when cetuximab was combined with afatinib (Janjigian *et al*, 2014). Our previous studies showed that treatments of erlotinib-resistant models with an anti-EGFR antibody evoke a compensatory response that up-regulates two EGFR family members, HER2 and HER3 (Mancini *et al*, 2015). Hence, we combined a TKI and mAbs against EGFR/HER-family members and observed effective inhibition of erlotinib-resistant tumors in cell line xenografts (Mancini *et al*, 2015; Mancini *et al*, 2018; Romaniello *et al*, 2018; Romaniello *et al*, 2020).

The studies presented herein assumed that upfront treatments of early tumors, harboring only one EGFR mutation, will prevent secondary resistance, thereby preempt sequential therapies using next-generation TKIs (aka, whack a mole protocols (Costa & Kobayashi, 2015)). To test this prediction, we applied combinations of two clinically approved mAbs, cetuximab and trastuzumab (an anti-HER2 mAb), together with one of the three TKIs approved for first-line treatment (e.g., erlotinib, afatinib, or osimertinib). In line with our prediction, when each TKI was applied on animal models harboring single-site EGFR mutants, all tumors were initially inhibited but rapidly relapsed. In contrast, therapeutic efficacy was strongly enhanced when each of the three TKIs was combined with the pair of antibodies. Aiming at the underlying molecular mechanisms, we found that TKI-mAbs combinations effectively block M phase, including cytokinesis, by inhibiting FOXM1, as well as instigate downregulation of several receptors for survival factors. Importantly, the TKI + mAbs combinations showed strong activity in two patient-derived xenograft (PDX) models, one carrying the prevalent exon 19 deletion and the other harboring the L858R-EGFR mutation. Taken together, the observations we made offer a potential first-line treatment strategy capable of preempting the recurring emergence of resistance to TKIs.

# Results

## In the 1st-line scenario, combining mAbs and TKIs decreases abundance of EGFR and other survival receptors, as well as elevates HER3 and blocks ERK and AKT

Because resistance to TKIs frequently associates with emergence of new mutations, whereas resistance to mAbs seldom engages mutations, combining TKIs and mAbs might delay resistance. To examine this prediction in the context of TKI-naïve tumors and upfront treatment strategies, we adopted two NSCLC models: PC9 cells, which express EGFRs harboring a short deletion in exon 19, and H3255 cells expressing the L858R mutation. As a prelude to animal studies, we separately combined *in vitro* mAbs against EGFR and HER2 (cetuximab and trastuzumab, respectively) and three different EGFR-specific TKIs, erlotinib, afatinib, and osimertinib, which respectively represent the first-, second-, and third-generation inhibitors. In both models, treatment with each TKI strongly inhibited phosphorylation of EGFR and HER2, as well as inactivated ERK and AKT (Fig 1A). By contrast, the mixture of two mAbs only

moderately inhibited phosphorylation of EGFR, ERK, and AKT, but the mAbs downregulated both EGFR and HER2, especially when combined with TKIs. This is in accordance with the ability of the mAbs to induce receptor internalization and degradation (Ben-Kasus *et al*, 2009; Spangler *et al*, 2010). Next, we asked if clearance of receptor tyrosine kinases (RTKs) extends to additional receptors for survival factors, which have previously been implicated in resistance to TKIs (Bean *et al*, 2007; Engelman *et al*, 2007). A clear reduction in the abundance of the receptors for the hepatocyte growth factor (MET), GAS6 (AXL) and the insulin-like growth factor 1 (IGF1R) was observed when either cell line was exposed to TKI + 2XmAbs. Although the mechanisms underlying trans-downregulation of other RTKs remain unclear, they might explain drug interactions. For instance, suppression of non-targeted RTKs may be mediated by heterodimer formation between EGFR and either MET, IGF1R or AXL (Jo *et al*, 2000; Balana *et al*, 2001; Brand *et al*, 2017). Interestingly, all drugs elevated abundance of un-phosphorylated HER3 molecules. Flow cytometry enabled us to focus on the surface-localized HER3. This confirmed downregulation of EGFR by all treatments that used 2XmAbs, and indicated that at least a fraction of the up-regulated HER3 molecules localized to the cell surface (Fig 1B). As a complementary assay, we used confocal microscopy (Fig 1C), which validated that EGFR and HER2 were downregulated following treatment with 2XmAbs and the TKIs. Notably, the immunofluorescence analysis confirmed drug-induced up-regulation of surface HER3, as well as revealed an increased intracellular pool, in line with a previous report (Sergina *et al*, 2007). In conclusion, concurrent treatments with three versions of TKI + 2XmAbs suppressed EGFR, HER2, and downstream signaling pathways. Along with the targeted receptors, we observed downregulation of additional RTKs, but all treatments associated with increased levels of HER3.

## Combinations of 2XmAbs and TKIs reduce cell viability, arrest cell cycle progression, and increase apoptosis

To examine effects on viability, we treated PC9 cells for 72 h with 2XmAbs in combination with low or high concentrations of EGFR-specific TKIs and assayed the conversion by live cells of MTT (3-(4,5-dimethylthiazol-2-yl)-2,5-diphenyltetrazolium bromide) to an insoluble formazan (Fig EV1A). When combined with 2XmAbs and used at lower concentrations, all three TKIs cooperatively inhibited cell viability, relative to the respective monotherapies. However, this effect disappeared when we used higher TKI concentrations. Similar effects were observed when we applied the same assay on H3255 cells (L858R-EGFR: Fig EV2A). Remarkably, however, when singly applied on H3255 cells, 2XmAbs achieved a much stronger inhibitory effect, 40%, as compared to < 10% in PC9 cells.

Probing cell extracts revealed that 2XmAbs strongly inhibited pERK in both cell lines, but the antibodies almost completely erased the pAKT signal only in H3255 cells (Fig EV2B). These observations explain the stronger effect of 2XmAbs on viability of H3255 cells. Interestingly, although we observed reduced phosphorylation of EGFR and HER2 in response to TKIs, 2XmAbs weakly enhanced phosphorylation signals, probably due to short-term agonistic effects of bivalent antibodies. Next, we assayed apoptosis in PC9 (Fig EV1B) and H3255 cells (Fig EV2C). Combining 2XmAbs with either erlotinib or osimertinib, at relatively low concentrations

(10–40 nM), resulted in elevated expression of two markers of cell death, BIM and cleaved caspase-3 (CC3), along with downregulation of survivin, an anti-apoptosis protein. In addition, we observed upregulation of γH2AX in cells treated with the drug combinations. Notably, when applied alone all three TKIs induced relatively strong apoptosis, whereas 2XmAbs only weakly regulated the four markers of apoptosis. In line with this, flow cytometry (Fig EV1C) and CC3 immunofluorescence of both PC9 and H3255 cells (Appendix Figs. S1A-S1D) hardly detected mAb-induced apoptosis,

but all three TKIs increased the fractions of cells undergoing apoptosis. These fractions were further increased when each TKI was combined with 2XmAbs.

To further evaluate the ability of the drug combinations to regulate cell cycle progression, we applied mass cytometry (CyTOF), a technique permitting simultaneous probing with 5-iodo-2-deoxyuridine, which marks cells in S phase, and antibodies against cyclin B1, phosphorylated retinoblastoma protein, and phosphorylated histone H3 (Behbehani *et al*, 2012). When singly applied, 2XmAbs

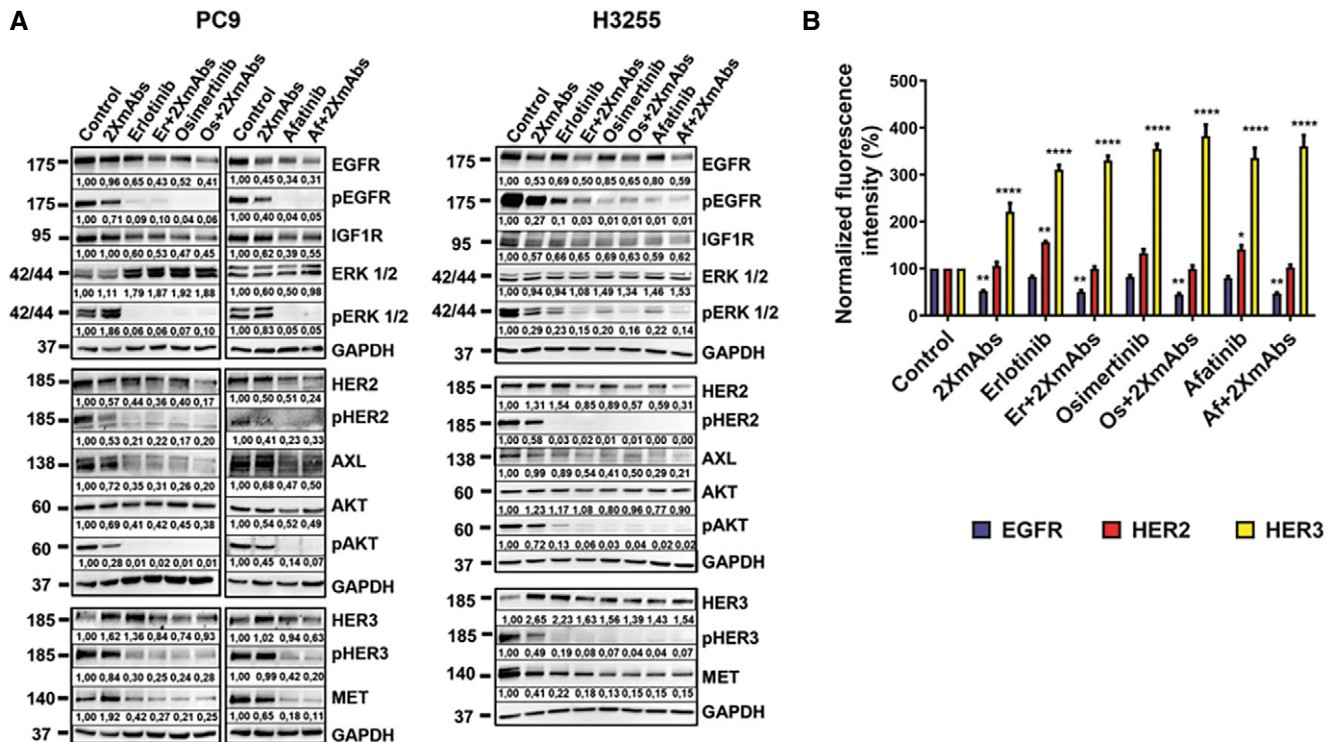

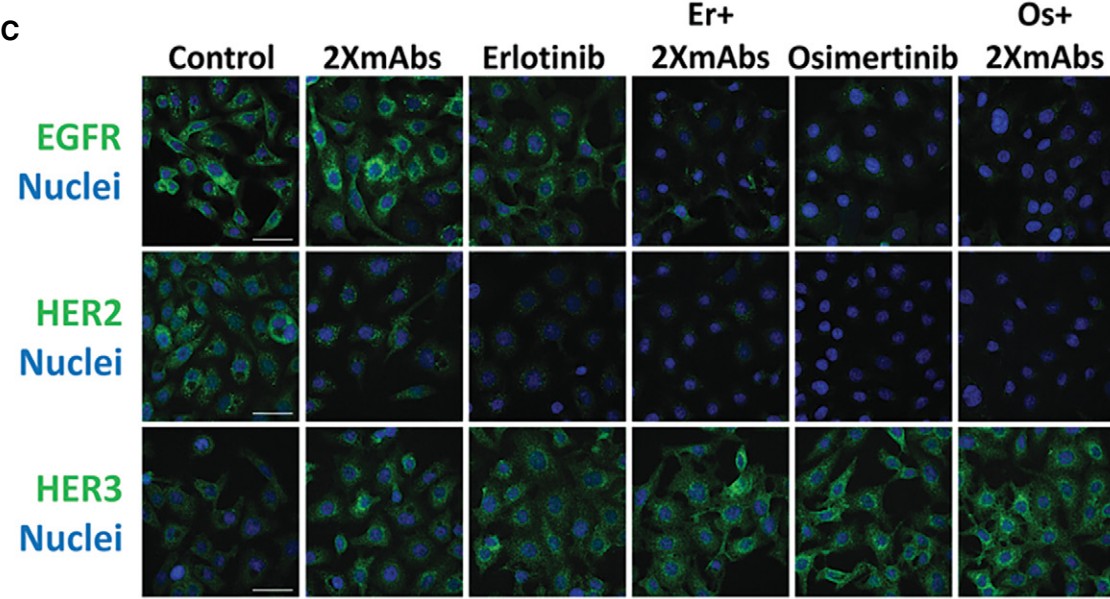

**Figure 1.**

**Figure 1. Combinations of mAbs and EGFR-specific TKIs downregulate *in vitro* several receptors for survival factors and effectively block the ERK and AKT pathways.**

A  NSCLC expressing single-site mutants of EGFR, PC9 ($3 \times 10^6$) or H3255 ($8 \times 10^6$), were seeded in 10-cm dishes. On the next day, complete media were replaced with media containing serum (1%) and the cells were treated for 24 h with different EGFR-specific TKIs (erlotinib, 50 nM; osimertinib, 50 nM, or afatinib, 10 nM), either alone or in combination with 2XmAbs (cetuximab and trastuzumab, 5 μg/ml each). Thereafter, cells were washed with cold saline and extracted. Proteins were separated using gel electrophoresis and transferred onto nitrocellulose membranes. After blocking, membranes were incubated overnight with the indicated primary antibodies, followed by incubation with horseradish peroxidase-conjugated secondary antibodies (60 min), and treatment with Clarity™ Western ECL Blotting Substrates (Bio-Rad). ECL signals were detected using the ChemiDoc™ Imaging System (Bio-Rad) and images were acquired using the ImageLab software. Signals (relative to Control) were quantified and normalized to the signals of GAPDH (numbers shown below each lane).

B  PC9 cells ($1 \times 10^6$) were seeded in 6-well plates and treated as in (A). After washing with acidic buffer (glycine 100 mM, pH 3.0), cells were incubated with fluorescently labeled antibodies against EGFR, HER2, and HER3, and surface expression levels of each receptor were analyzed using flow cytometry. The normalized fluorescence intensity is shown as averages + SEM of four experiments. Significance was assessed using two-way ANOVA followed by Dunnett's multiple comparisons test. Note that non-significant comparisons are not shown. See *P*-values in Appendix Table S2.

C  PC9 cells were seeded on coverslips and treated for 24 h as in (A). Cells were washed in acidic buffer, fixed in paraformaldehyde (4%), and incubated with specific primary antibodies, followed by an Alexa Fluor 555-conjugated secondary antibody (pseudo colored in green). DAPI (blue) was used to stain nuclei. Images were captured using a confocal microscope (40× magnification). Scale bars, 40 μm.

Source data are available online for this figure.

weakly arrested cells at G0/G1, but much larger effects were caused by the TKIs, which increased and decreased the G0 and M fractions, respectively, as well as reduced the numbers of Ki67-positive cells (Appendix Figs. S2A-S2E). Taken together, our results portray 2XmAbs as an enhancer of the effects induced by various TKIs on viability, apoptosis, and growth arrest of NSCLC cells expressing single-site EGFR mutants (T790WT).

**Neither erlotinib nor 2XmAbs can prevent tumor regrowth, but the combination prevents relapses of a T790WT animal model**

Since combinations of 2XmAbs and TKIs displayed cooperativity *in vitro*, we predicted pharmacological cooperation in animal models. Hence, we implanted PC9 cells (E746_A750 del-EGFR) in the flanks of immunocompromised mice and once tumors became palpable, mice were randomized to groups that were treated with TKIs (daily, oral gavage), 2XmAbs (twice weekly, intraperitoneal injections), or different combinations of the two treatments. All treatments were stopped after 54 days (or earlier), but we kept monitoring animals for 4 additional months. Appendix Figure S3A presents the averages of tumor volumes, and Appendix Figure S3B presents animal survival. The final measurements of tumor volumes are shown in Appendix Fig S3C. Curves corresponding to individual mice are shown in Appendix Fig S3D. All mice treated with afatinib or erlotinib monotherapy displayed short regressions followed by rapid progressions. The majority of mice treated with osimertinib or 2XmAbs initially regressed but relapsed later. In line with drug cooperation, both erlotinib + 2XmAbs and osimertinib + 2XmAbs fully inhibited tumors as long as treatment continued. Furthermore, following cessation of treatment we observed slow relapses in only 3 of 9 mice (33%) of the erlotinib + 2XmAbs group and in all but one animal (89%) of the osimertinib + 2XmAbs group. In conclusion, combining 2XmAbs and a TKI completely prevented relapse of a T790WT xenograft model, as long as mice were under treatment. This contrasted with the mostly short-term inhibitory effects achieved by either 2XmAbs or the TKIs.

We noted that afatinib showed no effect, probably because of the low dose we applied (2.5 mg/kg) due to the relatively high toxicity of this inhibitor (Ninomiya *et al*, 2013; Yu *et al*, 2018). Hence, we next examined TKI dosage effects. In addition, the new experiment applied longer treatments (90 instead of 54 days). Note, however, that for technical reasons we reduced the frequency of TKI

administration in the last month. The results presented in Fig 2AandB indicated that almost all mice treated with either erlotinib (50 mg/kg) or 2XmAbs experienced initial tumor regression followed by rapid progression. This was also the case when 2XmAbs was combined with relatively low doses of erlotinib. In stark contrast, all mice treated with the combination of 2XmAbs and high-dose erlotinib were apparently cured (note that one animal was lost in the course of this long experiment). Specifically, no relapses were observed during the three months of continuous treatment and > 110 days after treatment ending (see statistical analyses of tumor volumes in Fig 2C and tumor growth curves per each animal in Fig 2D). In addition, all treatments were associated with no apparent toxicity, as revealed by following animal body weight (Appendix Fig S7). In conclusion, the 1st-line treatment scenario revealed striking drug cooperation that depends on a critical threshold dose of the TKI, as well as the length of continual treatment with drug combinations.

**Along with treatment duration and TKI dose, high specificity to EGFR is critical for cooperation with antibodies**

The above-described setting of animal experiments, 90 days of treatment followed by 120 days of drug holiday, was employed when testing additional TKIs: osimertinib, afatinib, and a negative control TKI, imatinib, which is specific to BCR-ABL. Consistent with lack of specificity to EGFR, imatinib showed no effects on tumor growth and no synergy when combined with 2XmAbs (Fig 3A and B). In contrast, the same figures show that osimertinib and afatinib displayed strong cooperativity and no detectable toxicity (Appendix Fig S7) when combined with 2XmAbs. Note that Fig 3C presents the final tumor volumes, whereas Fig 3D shows growth curves corresponding to individual tumors and mice. Because both osimertinib and erlotinib were superior to afatinib, we performed a head-to-head comparative study that examined combinations with 2XmAbs and relatively short treatments (30 days; Fig EV3A and B). Note that final tumor volumes are presented in Fig EV3C, and tumor growth curves corresponding to individual mice are shown in Fig EV3D. Interestingly, similar rates of cure were observed, but the rates were lower than the scores achieved when mice were treated for longer time (90 days). In conclusion, combining mAbs and a TKI can achieve strong drug cooperativity, which requires high TKI dose and specificity, along with prolonged duration of treatment.

## Animal treatment with different TKI + 2XmAbs combinations induces downregulation of survivin and reduces abundance of several receptors for survival factors

Next, we addressed in animals the mechanisms underlying the cooperative effects of antibodies and TKIs. PC9 cells were implanted in the flanks of immunocompromised mice (2–3 animals per group).

Treatment lasted only one week and included 2XmAbs, TKIs, or the corresponding drug combinations. Note that we applied afatinib and erlotinib at the commonly used doses (5 and 50 mg/kg, respectively). However, due to its higher efficacy and relatively low toxicity in our models, osimertinib was applied at 5 mg/kg (or 10 mg/kg in other experiments). Tumor volumes were measured and presented in Fig 4A. Thereafter, all mice were sacrificed and tumor

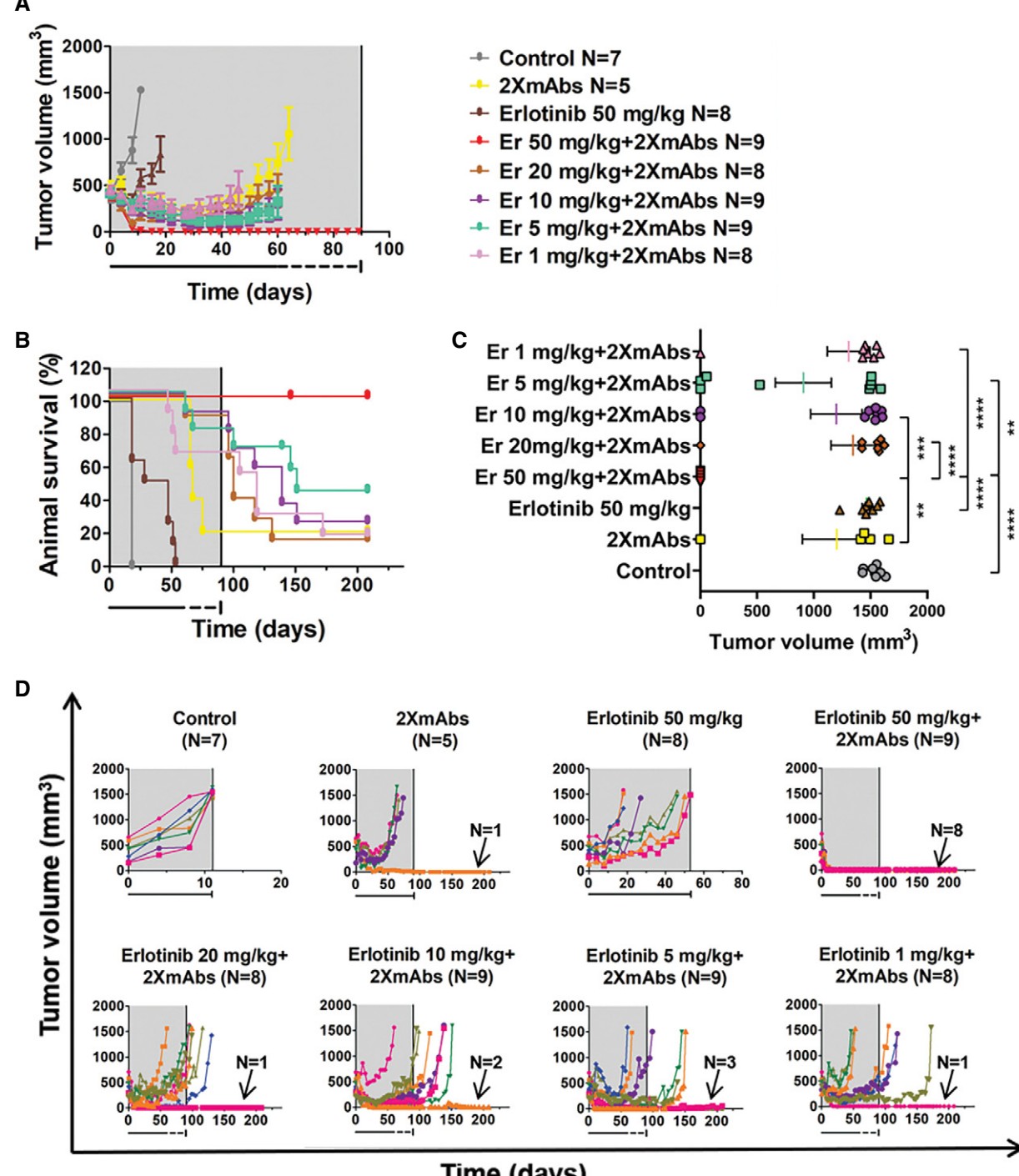

**Figure 2.**

**Figure 2.  Neither erlotinib nor 2XmAbs are effective, but the combination cures a xenograft model driven by a single-mutation EGFR.**

PC9 cells ($3 \times 10^6$/mouse) were subcutaneously implanted in the flanks of CD1-nu/nu mice. When tumors became palpable, mice were randomized in groups of 5–9 animals and treated for 90 days (gray areas) with 2XmAbs (cetuximab plus trastuzumab, 0.2 mg/mouse/injection), once every three days, or with erlotinib (50 mg/kg/day), once per day. Alternatively, mice were treated with combinations of erlotinib (50, 20, 10, 5, or 1 mg/kg) and the two monoclonal antibodies. Following 60 days of treatment, we reduced the frequency of erlotinib administration to once every other day (underneath dotted line).

A, B   Tumor volumes (A) and animal survival (B) are shown. Mice were euthanized when tumor size reached 1,500 mm$^3$. Data are means ± SEM from 5–9 mice per group.
C      Statistical analysis of tumor volumes corresponding to the last measurement for each mouse was performed using one-way ANOVA followed by Tukey's multiple comparison test. Results are shown as mean ± SEM. See *P*-values and number of mice per group (N) in Appendix Table S2. Note that only the significant comparisons are shown.
D      Shown are tumor volumes of individual mice of each group. Note that each animal is represented by a different color. The respective numbers of tumor-free mice are indicated. The gray areas represent treatment phases, and they are followed by drug holidays (blank).

extracts were analyzed using immunoblotting (Fig 4B). As expected, we observed some variation among the three mice of each group. Hence, we performed a reference animal study that combined imatinib, a BCR-ABL inhibitor, with 2XmAbs (Fig EV4). In addition, we confirmed, using immunoblotting of extracts derived from PC9 cells that imatinib was unable to inhibit EGFR phosphorylation (Fig EV4C).

Despite inter-animal variation, we concluded that the following biochemical features characterized tumors treated with all three variants of TKI + 2XmAbs, as compared to the respective monotherapies, or the 4th combination, imatinib + 2XmAbs:

1   Downregulation of EGFR and inhibition of EGFR phosphorylation were observed in mice treated with 2XmAbs or TKIs, respectively. These effects, however, were strongly enhanced in mice treated with each of the three TKI + 2XmAbs combinations.
2   In addition to HER2 and pHER2, all drug combinations associated with downregulation of IGF1R, AXL, and MET, but VEGFR2 remained almost unchanged. Similar alterations were detected *in vitro* (Fig 1A). However, unlike the *in vitro* observations, after one week of treatment with TKI + 2XmAbs, HER3 was also downregulated.
3   In line with downregulation of mutant EGFRs and several other RTKs, the combinations of 2XmAbs with either erlotinib, osimertinib, or afatinib led to inactivation of ERK and AKT. No similarly uniform inactivation of downstream signals was observed in mice treated with imatinib + 2XmAbs (Fig EV4B).
4   An antiapoptotic protein, survivin, was strongly expressed in the control tumors. This protein underwent downregulation following treatment with either 2XmAbs or TKIs, and it was completely absent in tumors treated with the three drug combinations (Fig 4B). Importantly, imatinib alone exerted no effect on survivin levels (Fig EV4B) or on tumor volume (Fig EV4A).

Notably, due to relatively high abundance of BIM and cleaved caspase-3 in the control mice, we were unable to detect alterations in these markers of apoptosis.

To complement the analyses shown in Fig 4, we assessed both apoptosis (Appendix Figs. S4A and S4B) and proliferation (Appendix Figs. S4C and S4D) within tumors. In line with the *in vitro* assays, imatinib was inactive, erlotinib clearly induced apoptosis, and 2XmAbs arrested proliferation, while erlotinib + 2XmAbs enhanced both apoptosis and growth arrest. In conclusion, short-term animal treatments with combinations of antibodies and EGFR-specific TKIs enhanced apoptosis, reduced cell proliferation, and caused downregulation of the antiapoptotic protein survivin, along with several receptors for growth factors. These observations might explain the drug cooperation we observed in mice engrafted with models of early EGFR-positive NSCLC.

**TKI plus mAb mixtures overcome in animals the commonly observed acquired resistance to first-line kinase inhibitors**

Although several EGFR-specific TKIs have been approved as first-line treatments, osimertinib showed superior efficacy with a similar safety profile, relative to erlotinib and gefitinib (Soria *et al*, 2018). However, resistance to upfront osimertinib is inevitable and might involve amplification of *MET* or *EGFR*, mutations in *KRAS*, *MEK1*, and *PIK3CA*, along with the EGFR-C797S mutation (Ramalingam *et al*, 2018). Currently, it is unclear how to treat tumors that evolved resistance to first-line osimertinib. Similarly, it is still debated which is the TKI of choice, and whether sequential regimens would be beneficial. In light of the strong anti-tumor activity displayed by combinations comprising TKIs and 2XmAbs, we simulated in animals the evolvement of resistance to 1st-line treatments. PC9 cells were injected into the flanks of CD1 nude mice, and when the tumors reached 500 mm$^3$, mice were daily treated with either

**Figure 3.  A combination of mAbs specific to EGFR and to HER2 collaborates with both second- and third-generation EGFR TKIs, but a BCR-ABL TKI displays no cooperative effects.**

PC9 cells (exon 19 deletion) were subcutaneously implanted in the flank of CD1-nu/nu mice ($3 \times 10^6$/mouse). When tumors became palpable, mice were randomized in groups of 5–9 animals and treated for 90 days (gray areas) with 2XmAbs (cetuximab plus trastuzumab, each at 0.1 mg/mouse/injection) once every three days, or daily with different TKIs: osimertinib (5 mg/kg), afatinib (5 mg/kg), or imatinib (100 mg/kg), either alone or in combination with the two antibodies. Following 60 days of treatment, the frequency of TKI administration was reduced to once every other day (underneath dotted lines), while mAb treatment remained unaltered.

A, B   Tumor growth (A) and animal survival (B) are shown. Mice were euthanized when tumor size reached 1,500 mm$^3$. Data are means ± SEM from 5–9 mice of each group.
C      Statistical analysis of tumor volumes corresponding to the last measurement for each mouse was performed using one-way ANOVA followed by Tukey's multiple comparison test. Results are shown as mean ± SEM. See *P*-values and number of mice per group (N) in Appendix Table S2. Note that only the significant comparisons are shown.
D      Shown are tumor volumes corresponding to individual animals of each group. The numbers of mice with undetectable tumors are indicated. Note that the control and 2XmAbs arms are shared with Fig 2. The gray areas represent treatment phases, and they are followed by drug holidays (blank).

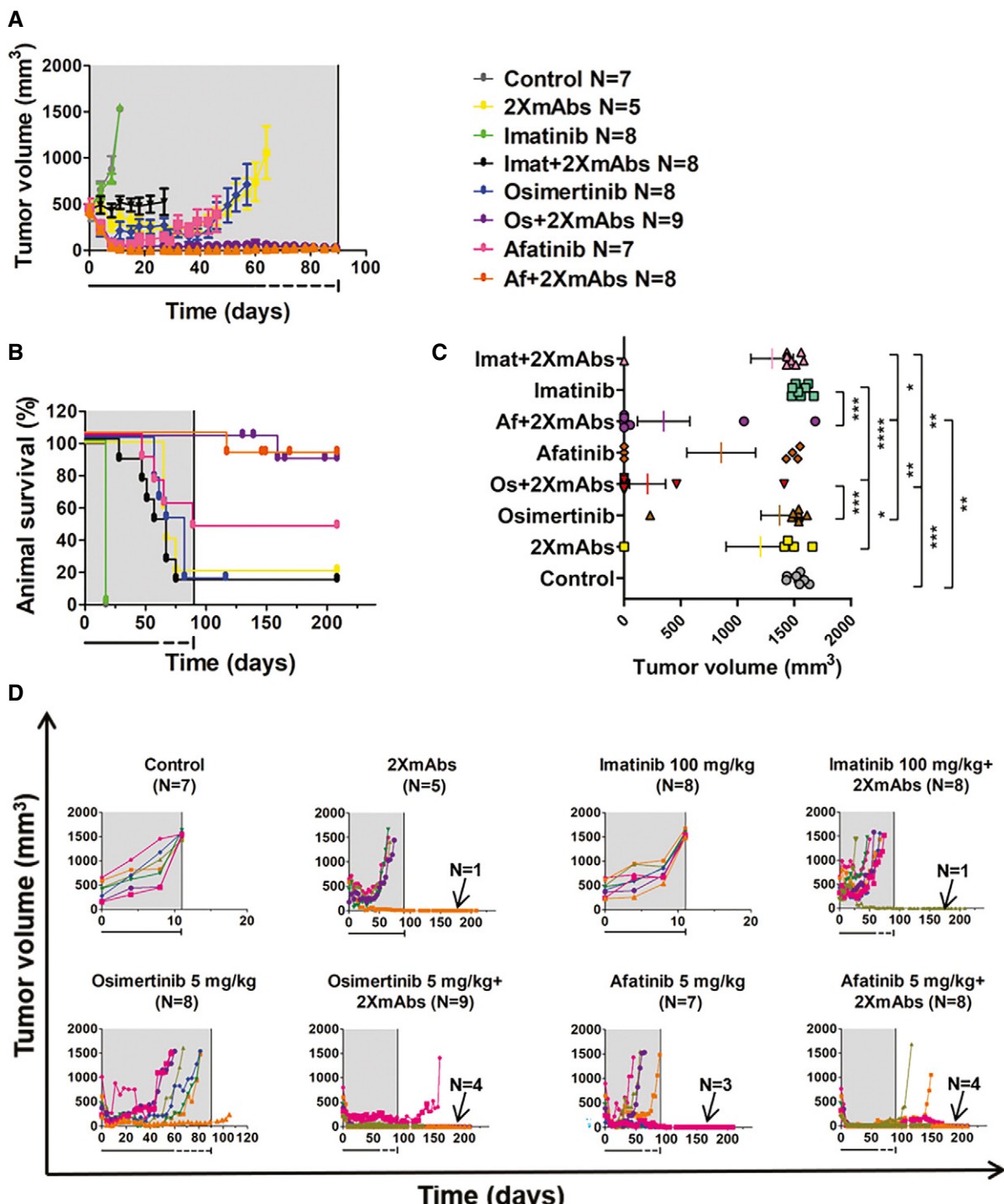

**Figure 3.**

erlotinib (50 mg/kg; Fig 5A) or osimertinib (5 mg/kg; Fig 5B). Under these conditions, most tumors exhibited regression, followed by rapid regrowth. Once each relapsing tumor reached 800 mm³, we administered the mixture of cetuximab and trastuzumab, and continued oral delivery of the respective TKI. Remarkably, all tumors treated with the combination of TKIs plus 2XmAbs rapidly regressed. Furthermore, in the majority of animals we observed complete tumor disappearance. In marked contrast, when mice with TKI-resistant tumors were sequentially treated with only 2XmAbs, after tumors became resistant to erlotinib (Appendix Fig S5A), or to osimertinib (Appendix Fig S5B), the responses to the antibodies widely varied: a few tumors responded well and eventually disappeared, while others kept progressing under treatment. In conclusion, the results presented in Fig 5 and Appendix Figure S5

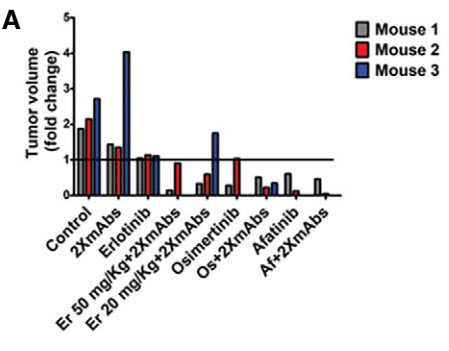

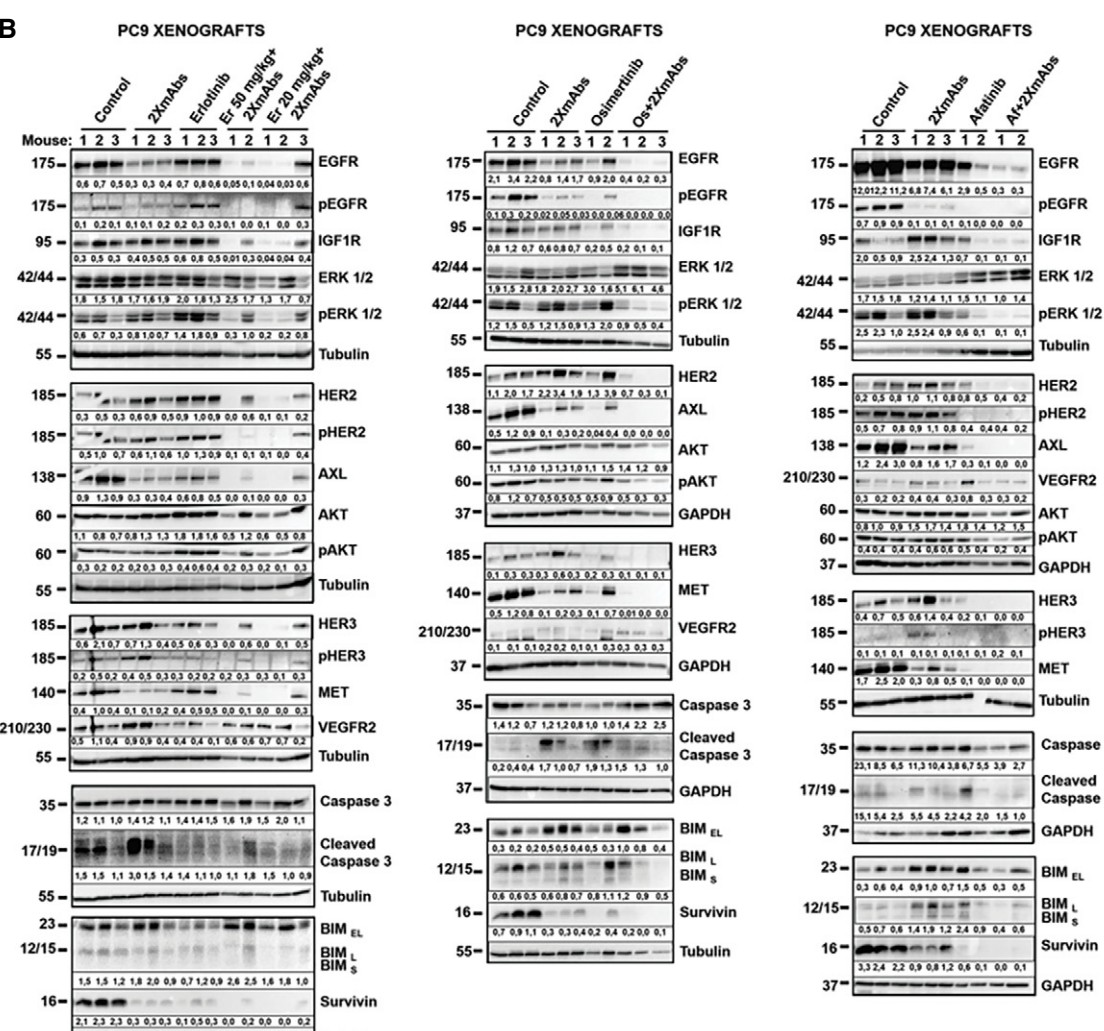

**Figure 4.  Animal treatments with combinations of 2XmAbs plus EGFR-specific TKIs induce downregulation of survivin and several receptors for growth/survival factors.**

CD1-nu/nu mice carrying PC9 xenografts were divided in groups (2–3 mice/group) and treated for seven days with different TKIs: erlotinib (50 mg/kg or 20 mg/kg), osimertinib (5 mg/kg), or afatinib (5 mg/kg), either alone or in combination with 2XmAbs (cetuximab and trastuzumab, each at 0.1 mg/mouse/injection). Note that when singly applied, we used erlotinib at 50 mg/kg.

A   Shown are fold changes in tumor volume at day 7 of treatment. The horizontal line indicates no change in tumor volume (fold change) between days 0 and 7 of treatment.

B   After treatment, all mice were sacrificed and tumors extracted. Protein extracts were resolved by means of electrophoresis and transfer to nitrocellulose membranes, which were later incubated overnight with the indicated antibodies. This was followed by incubation with peroxidase-conjugated secondary antibodies. Signals were detected using Chemidoc™ (from Bio-Rad), quantified and normalized to the signals of GAPDH or tubulin (numbers shown below each lane).

Source data are available online for this figure.

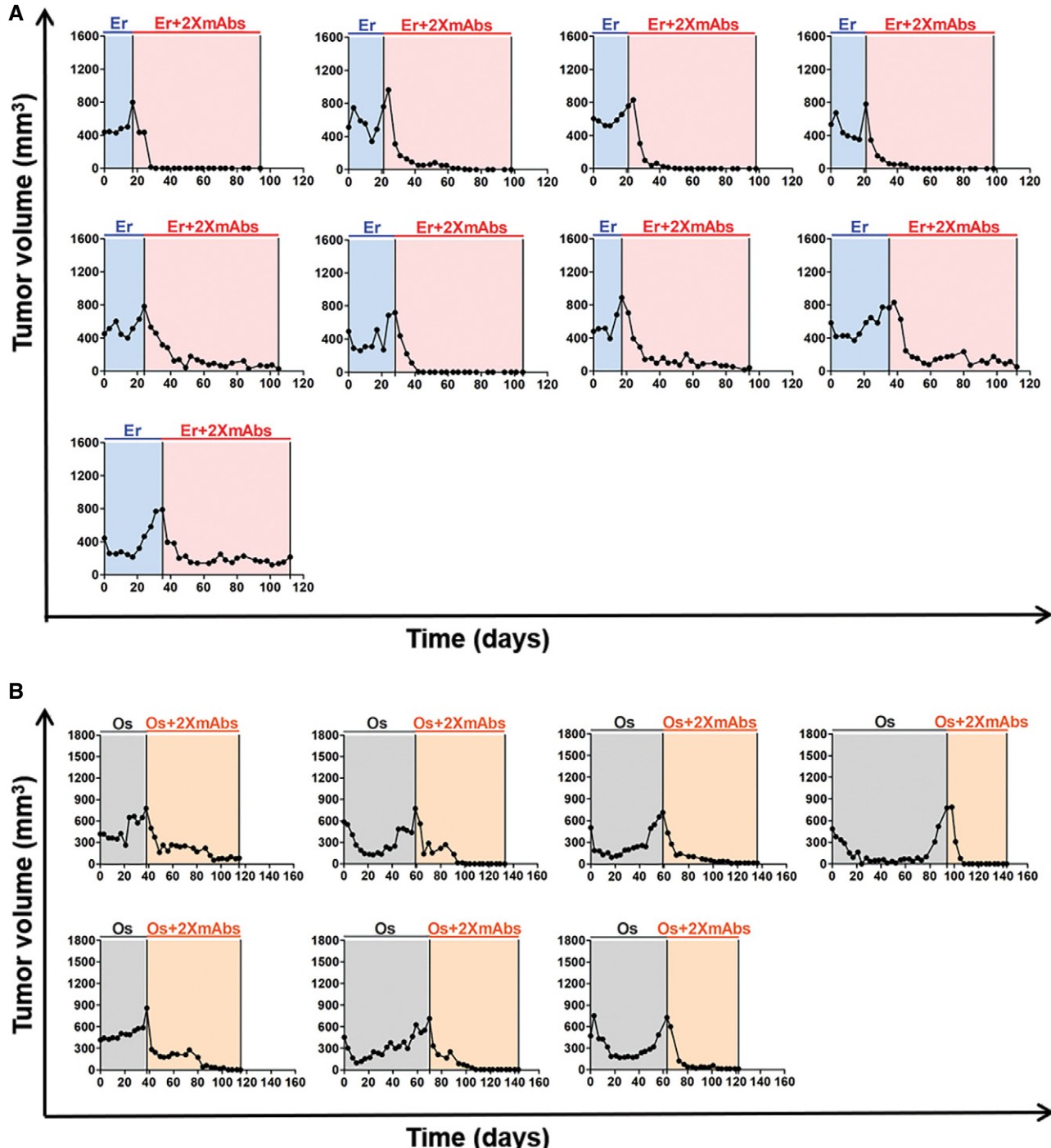

**Figure 5. Combination treatments comprising a mixture of two antibodies (cetuximab and trastuzumab) and either erlotinib or osimertinib overcome resistance to TKI in an animal model.**

A, B   PC9 cells ($3 \times 10^6$/mouse) were subcutaneously injected in the flanks of CD1-nu/nu mice. When tumors reached a volume of approximately 500 mm³, mice were treated daily with erlotinib (50 mg/kg, blue area; panel A) or with osimertinib (5 mg/kg, gray area; panel B) using oral gavage. Initially, all tumors displayed stable disease or they regressed, but eventually all started relapsing. Once relapsing tumors reached 800 mm³, while under erlotinib or osimertinib treatment, we supplemented the treatment with a combination of 2XmAbs (cetuximab and trastuzumab, each at 0.1 mg/mouse/injection; twice a week) plus either erlotinib (50 mg/kg; A, red area) or osimertinib (5 mg/kg; B, orange area). Tumor volumes of individual mice are shown.

reinforce the strongly cooperative mode of interactions between 2XmAbs and TKIs, and they validate that relatively aggressive tumors, which acquired resistance to TKIs, are still controllable by two different TKI + 2XmAbs combinations.

## Complete responses of patient-derived xenografts corresponding to prevalent EGFR mutations can be achieved by upfront combination treatments

Patient-derived tumor xenografts (PDX) have emerged as a powerful technology, capable of retaining the clonal and mutational heterogeneity of the original specimens. To test the efficacy of combining EGFR-specific TKIs and 2XmAbs (cetuximab and trastuzumab) in such genetically more heterogeneous model systems, we made use of two PDX models of NSCLC, both from the Jackson Laboratory: TM00199, expressing L858R-EGFR, and TM00193, expressing E746_A750 del-EGFR. Immunocompromised mice bearing tumors expressing either the L858R mutation (Fig 6A) or the exon 19 deletion (Fig 6B) were treated for 32 or 42 days, respectively, with an EGFR-TKI, either erlotinib (50 mg/kg) or osimertinib (10 mg/kg). Alternatively, mice were treated with 2XmAbs (cetuximab + trastuzumab, each at 100 μg/injection), or with a combination of the two mAbs and a TKI. Although the TM00199 model displayed high responsiveness to TKIs, tumor growth rapidly resumed after treatment cessation (Fig 6A). Surprisingly, this model responded extremely well to the mixture of two antibodies: pre-established tumors rapidly regressed after initiation of treatment with 2XmAbs, and no relapses occurred for at least 2 additional months of drug holiday. Accordingly, mice treated with either combination of mAbs and TKIs displayed durable suppression of relapses following a short treatment (32 days). In fact, only one of 16 animals pre-treated with TKI + 2XmAbs relapsed during a long (>70 days) drug holiday.

Similar to observations made with cell line xenografts, the TM00193 model (expressing E746_A750 del-EGFR) initially responded to the monotherapies we applied (osimertinib, erlotinib, or 2XmAbs). However, relapses initiated thereafter and their growth rate accelerated after we stopped all treatments (Fig 6B). Importantly, both drug combinations (2XmAbs + erlotinib and 2XmAbs + osimertinib) achieved nearly complete tumor inhibition and no relapses were observed in the groups of 7–9 mice, for at least 25 days after treatment ending. In summary, by employing PDX models that retain the natural heterogeneity of lung cancer and represent the two most prevalent EGFR mutations, we validated the cooperative interactions between TKIs and mAbs, as well as demonstrated that this treatment scenario is able to durably control tumors expressing mutant forms of EGFR.

## Residual disease remaining after treatment of a PDX model with mAbs + TKIs displays sensitivity to re-application of the drug combination

Currently available data propose that drug-resistant cells may both pre-exist and evolve from drug-tolerant cells. For example, acquired resistance caused by the T790M mutation may occur either by selection of pre-existing T790M-positive clones or via evolution of initially T790WT clones (Hata *et al*, 2016). By focusing on erlotinib-resistant colonies that arose from a single, EGFR-addicted lung

cancer cell, another study concluded that drug-tolerant "persisters" may serve as latent reservoirs for the emergence of resistance mechanisms (Ramirez *et al*, 2016). Dissimilar routes might underlie emergence of resistance to TKI monotherapy and resistance to the combinations of mAbs and TKIs. The availability of PDX tumors (TM00193) that regressed after 42 days of treatment with erlotinib + 2XmAbs and re-emerged following a long drug holiday (3–4 months; see Fig 6B) permitted us to address dependence of the relapsing tumors on EGFR and HER2. To this end, we selected six of the relapsing animals and re-treated them with erlotinib + 2XmAbs. As shown in Fig 6C, all six relapsing tumors exhibited rapid regression, although none completely disappeared. These observations indicate that the residual disease that seeded them remained addicted, at least in part, to EGFR and/or HER2. Conceivably, in addition to rapid inhibition of pre-established tumors and prevention of relapses, strategies admixing TKIs and mAbs may confer long-term protection from relapses and, if re-applied after a prolonged drug holiday, they might control any remaining disease.

## Blocking pathways controlling the cell cycle underlies the ability of TKI + 2XmAbs to overcome drug resistance

In light of the ability of TKI + 2XmAbs to overcome drug resistance in animal models, the underlying mechanisms are of interest. Mice harboring pre-established PC9 xenografts were initially treated with erlotinib, until onset of resistance (gray area; see scheme in Fig 7A). Afterward, mice were switched to a combination of the TKI and two mAbs (cetuximab and trastuzumab; pink area in Fig 7A). Four groups of animals were sequentially sacrificed: Control group (pre-treatment; orange star, $N = 5$), Erlotinib-responding group (green, $N = 3$); Erlotinib-resistant group (blue, $N = 3$), and an Erlotinib + 2XmAbs-responding group (purple, $N = 2$). Comparison of pre-treatment tumor volumes and sizes at the time each mouse was sacrificed is shown in Fig EV5A. Next, RNA was extracted from each tumor. One portion was used for sequencing of the region corresponding to exons 19 and 20. While this analysis confirmed the del746-750 mutation, it detected no secondary mutations, such as T790M or C797S. The other portion of RNA underwent full sequencing. The heat map presented in Fig 7B lists genes that were differentially expressed (DE) between the Erlotinib + 2XmAb group and the Control group. Pathway enrichment analysis (Fig EV5B) linked the majority of genes downregulated by the drug combination to the cell cycle. This included processes controlled by FOXM1, polo-like kinase 1, oncostatin M, and aurora B. Using RT–PCR, we validated up-regulation of some of the altered genes (Fig EV5C). Next, we extracted each tumor and subjected the extracts to immunoblotting (Fig 7C and Appendix Figure S6). The results confirmed inhibition of EGFR and HER2, along with partial or complete downregulation of IGF1R, AXL, HER3, and MET in tumors responding to erlotinib + 2XmAbs. Concordant with RNA-seq and RT–PCR, several other proteins underwent collective up- or downregulation in all mice. For example, survivin (BIRC5), an inhibitor of apoptosis, was highly expressed in all five control tumors, downregulated in tumors responding to erlotinib, to be again up-regulated in all three tumors that progressed (Fig 7C). Finally, treatment with TKI + 2XmAbs completely erased survivin expression. Likewise, aurora kinase A (AURKA), TPX2, cyclin B1 (CCNB1), PRC1 (protein regulating cytokinesis), and KIF4a displayed partial downregulation

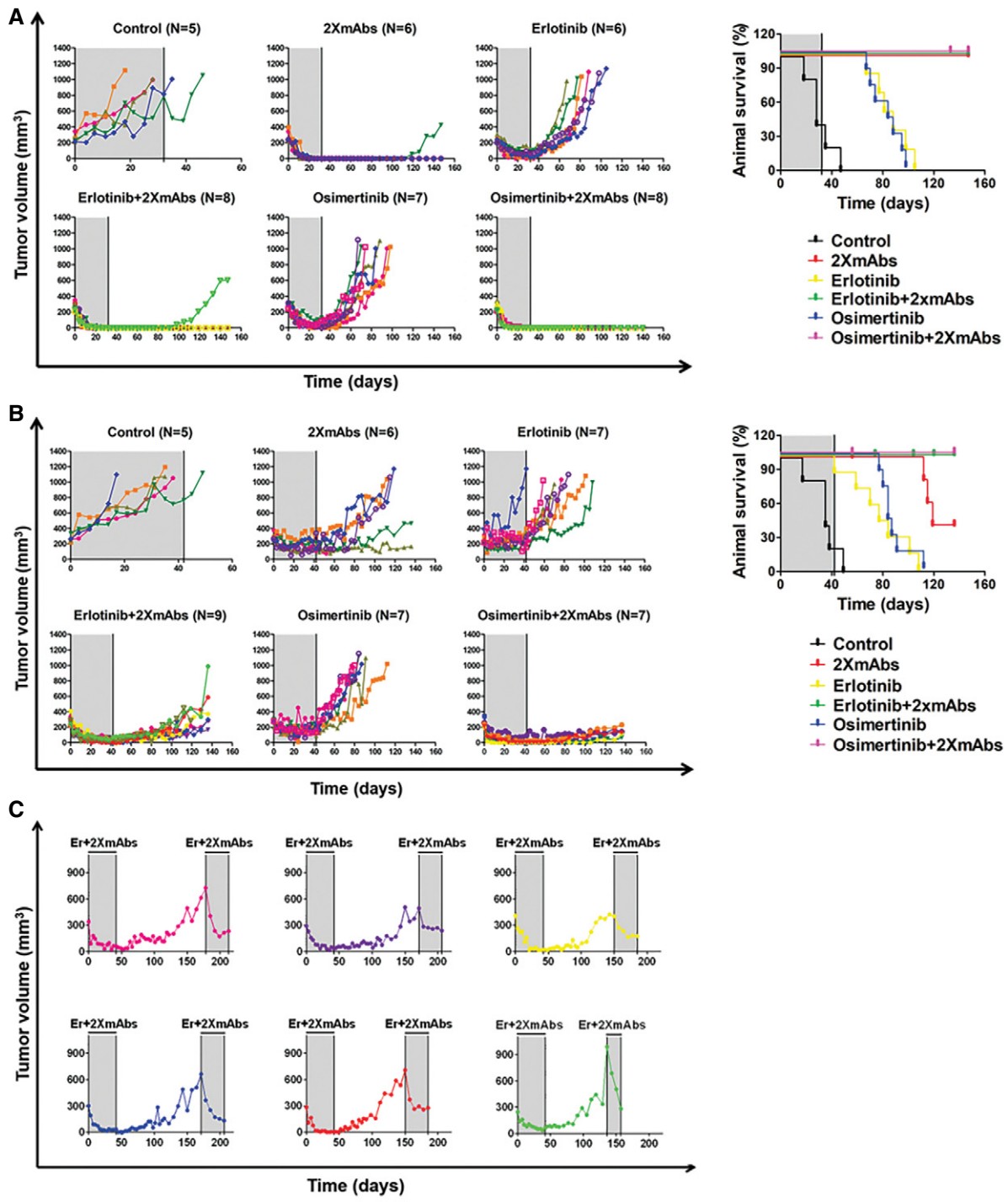

**Figure 6. PDX models corresponding to the most frequent founder EGFR mutations can be effectively inhibited by combinations of antibodies and TKIs; relapsing tumors remain sensitive to erlotinib + 2XmAbs even after a long drug holiday (> 3 months).**

A, B NSG mice were pre-implanted with tumor fragments derived from two different PDX models: (A) TM00199 (L858R-EGFR; from The Jackson Laboratory) or (B) TM00193 (E746_A750 del-EGFR; from The Jackson Laboratory). Once tumors reached approximately 300 mm³, mice were treated with monoclonal antibodies (2XmAbs, cetuximab plus trastuzumab, 200 μg/injection) twice a week, or with EGFR TKIs, erlotinib (50 mg/kg) or osimertinib (10 mg/kg) daily. Alternatively, mice were treated with a combination of 2XmAbs and either erlotinib or osimertinib, for either 32 days (A) or 42 days (B). Tumor growth was monitored twice a week. Notice that the antibodies were injected into the peritoneum and the TKIs were delivered orally. The figure shows tumor volumes of individual mice and the respective survival curves (right hand panels). The gray areas mark time windows of animal treatment.

C Mice presented in the erlotinib + 2XmAbs group in B (left panel, bottom row) were re-treated with the same drug combination (2XmAbs plus erlotinib) after a holiday period (> 136 days), during which they received no treatment. Note that different colors identify individual mice from the corresponding panel of B.

in mice treated with erlotinib and complete eradication in mice responding to the drug combination. Importantly, the majority of the respective genes are transcriptional targets of the forkhead transcription factor called FOXM1. Accordingly, both FOXM1 and its phosphorylated form were fully inhibited in tumors responding to the TKI + mAbs combination (Fig 7C).

*FOXM1* is an oncogene involved in cancer progression (Borhani & Gartel, 2020), and its transcriptional targets directly control the cell cycle (Laoukili *et al*, 2007). FOXM1 is negatively regulated by FOXO3a and competes for the same target genes (Lam *et al*, 2013). In turn, FOXO3a is negatively regulated, via phosphorylation, by both AKT and ERK (Borhani & Gartel, 2020). This causes nuclear exclusion of FOXO3a and consequent up-regulation of FOXM1. Consistently, immunofluorescence analysis of PC9 cells treated with either erlotinib or erlotinib + 2XmAbs detected nearly exclusive nuclear localization of FOXO3a (Fig EV5D). Parallel immunoblotting

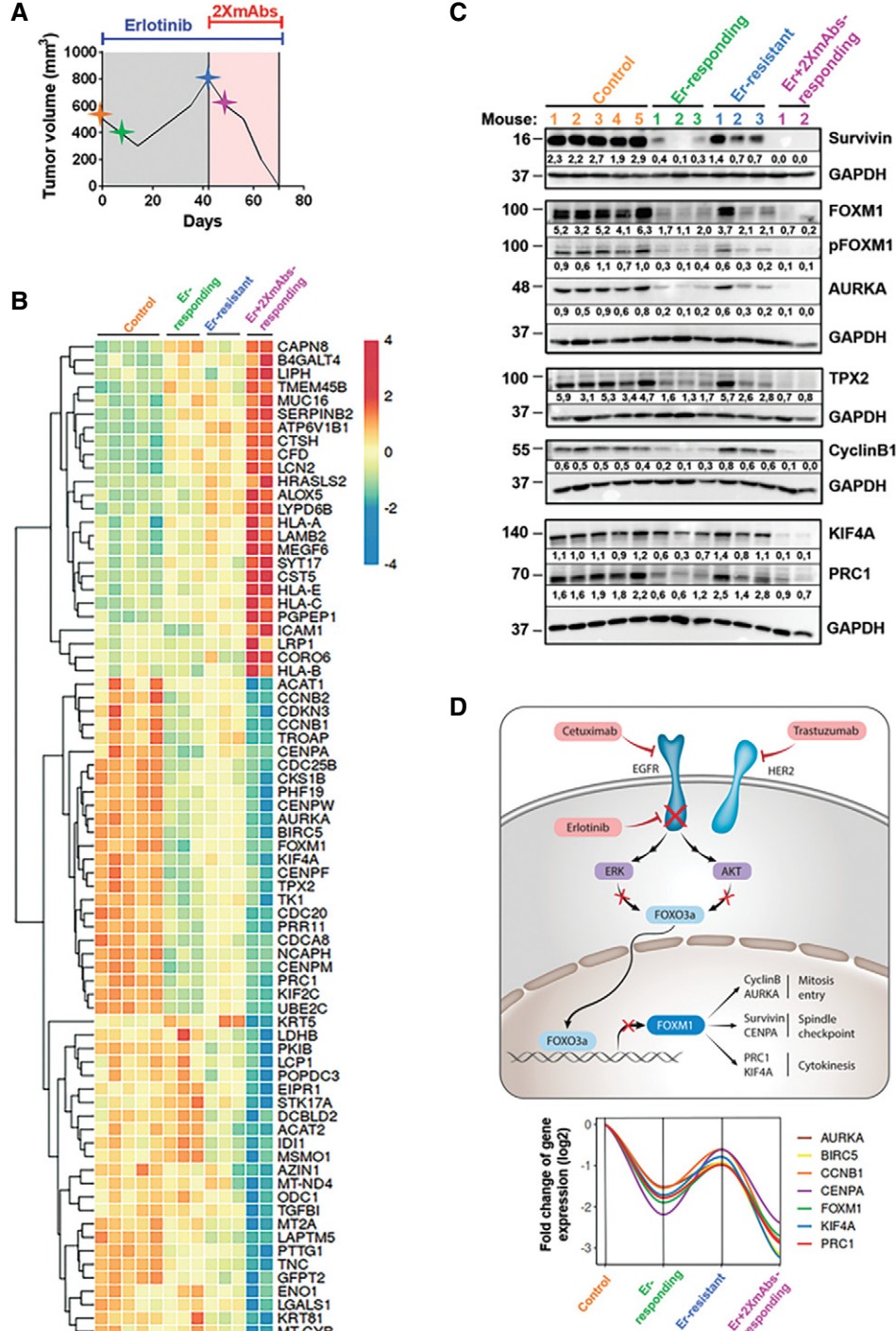

**Figure 7.**

**Figure 7. By blocking FOXM1 and pathways involved in mitosis and cytokinesis, a combination of two antibodies and erlotinib overcomes TKI resistance in an animal model.**

A  Scheme depicting the experimental plan and longitudinal sampling.

B  For RNA-seq analyses, PC9 cells ($3 \times 10^6$/mouse) were subcutaneously injected in the flanks of 13 CD1-nu/nu mice. When tumors reached 500 mm$^3$, mice were treated daily with erlotinib (50 mg/kg). Once tumors that initially responded to erlotinib monotherapy started relapsing and reached 800 mm$^3$, we switched to a combination of 2xmAbs plus erlotinib (50 mg/kg). The respective tumor volumes are presented in Fig EV5A. Control mice ($N = 5$) were sacrificed when tumor volumes reached approximately 500 mm$^3$ (orange star in A). Mice responding to erlotinib ($N = 3$) were sacrificed after one week of treatment (green star in A). Mice resistant to erlotinib ($N = 3$) were sacrificed when tumor volumes reached 800 mm$^3$ (blue star in A), whereas mice responding to erlotinib + 2xmAbs ($N = 2$) were sacrificed after one week of treatment (purple star in A). RNA was extracted from all 13 tumors and utilized for RNA-seq analysis. Differentially expressed (DE) genes in the erlotinib (Er) + 2xmAbs arm compared with the Control group are presented in the heat map.

C  Protein extracts were prepared from all tumors and analyzed using immunoblotting, as indicated. Signals were quantified and normalized to the signals of GAPDH (numbers shown below each lane).

D  Summary model: ERK and AKT phosphorylate the transcription factor FOXO3a, thereby dampen its ability to downregulate FOXM1. The combination of erlotinib, cetuximab, and trastuzumab overcomes TKI resistance by inhibiting ERK, AKT, and FOXM1 transcription. This downregulates several FOXM1 target genes, such as cyclinB1, AURKA, CENPA, BIRC5, PRC1, and KIF4A. All these genes play an essential role in the late steps of the cell cycle, primarily mitosis and cytokinesis, and their downregulation results in inhibition of cell proliferation. The lower panel refers to the tumor samples analyzed in B and shows the calculated fold changes of a gene module repressed after treatment with erlotinib + 2xmAbs.

Source data are available online for this figure.

confirmed downregulation of FOXM1, along with several target proteins, especially in cells treated with erlotinib + 2xmAbs (Fig EV5E). A summary model is presented in Fig 7D, along with the calculated fold changes of relevant transcripts (lower panel). According to the model, active EGFRs stimulate both ERK and AKT; hence, FOXO3a is inactivated and FOXM1 and several targets involved in cell cycle regulation are up-regulated. Once EGFR is inhibited, FOXO3a translocates to the nucleus and blocks FOXM1. As a consequence, a module of co-regulated genes undergoes concerted up-regulation in relapsing tumors and coordinated down-regulation in tumors responding to treatment. The module simultaneously regulates mitosis entry, the spindle checkpoint and, through genes like PRC1 and KIF4A, assembly of the spindle midzone (Subramanian *et al*, 2013). Because TKI + 2xmAbs more potently than either monotherapy represses this gene module, we raise the possibility that the cooperative effect of the drug combination (TKI + 2xmAbs) might be due to inhibition of cell cycle completion.

In summary, this study simulated in animals the situation of TKI-naïve NSCLC patients, who are subjected to first-line drug administration. The results of the simulations not only indicated superiority of all TKI + 2xmAbs combinations over the respective monotherapies, but also uncovered some features of the molecular mechanisms underlying the cooperative potency of admixing TKIs and antibodies: While antibody monotherapies only weakly inhibited cell proliferation, they critically enhanced the ability of TKIs to suppress transcriptional programs essential for cytokinesis and spindle assembly. Interestingly, this effect is complemented by the ability of TKI + 2xmAbs to downregulate several survival receptors. Hence, translation of the new lessons to clinical application might replace the widely used sequential treatments with next-generation inhibitors.

## Discussion

Although five clinically approved TKIs are currently available for treatment of patients with EGFR$^+$ NSCLC, almost all patients will eventually succumb to their disease. This is because the current clinical practice offers no solution beyond emergence of resistance to third-generation TKIs (Takeda & Nakagawa, 2019). For example, treatment with osimertinib provides significantly greater efficacy than chemotherapy in patients with T790M-positive tumors (Mok *et al*, 2017), but emergence of resistance is nearly inevitable and no solution is available also for the similarly large fraction of TKI-resistant patients whose tumors are T790WT. Potentially, immune checkpoint inhibitors might provide a solution. However, EGFR mutant tumors generally display low responses to immune checkpoint inhibitors (Hastings *et al*, 2019). Herein, we investigated in animal models an alternative treatment strategy. Previously, we treated NSCLC models expressing T790M-EGFR with anti-EGFR antibodies and observed up-regulation of HER2 (Mancini *et al*, 2015). Hence, we applied a combination with an anti-HER2 antibody. When tested on cell line xenografts of TKI-resistant models (T790M), we observed cooperative effects of the osimertinib + 2xmAbs combination (Romaniello *et al*, 2018). This prompted the current tests, which simulated *in vivo* treatments of TKI-naïve, T790WT NSCLC patients who undergo first-line drug administration.

Alongside regular xenografts and *in vitro* models, our study employed two patient-derived xenograft models of the most frequent EGFR mutations. To pre-clinically explore the full potential of combining antibodies and TKIs in the 1st-line treatment setting, we applied three TKIs currently approved for upfront treatments, namely erlotinib, afatinib, and osimertinib. Head-to-head animal comparative studies, which tested the three TKI + 2xmAbs combinations, concluded that all three scenarios are markedly more effective than the respective monotherapies, and they can persistently prevent relapses, not only of cell line xenografts but also patient-derived models. These observations, if confirmed and implemented in clinical practice of TKI-naïve patients, might preempt whack-a-mole sequential strategies, such as sequential afatinib and osimertinib (Hochmair *et al*, 2018). Notably, although all TKIs we tested as components of the upfront combination treatment have shown efficacy, imatinib, which ineffectively inhibits EGFR, could not complement the action of the antibodies. In addition to TKI specificity to EGFR, high dose of the inhibitor and prolonged treatments with the triplet appear essential for high efficacy.

Mechanisms underlying cooperative mAb-TKI interactions are of considerable importance. Pharmacological cooperation in terms of delaying onset of drug resistance might reflect the remarkably different mechanisms that confer resistance to the two classes of drugs: while both evoke compensatory bypass routes, genome-based

alterations versus immunological processes are the major drivers of resistance to TKIs and mAbs, respectively (Garraway & Janne, 2012; Mancini & Yarden, 2016; Marrocco et al, 2019). Only very few studies examined combinations of TKIs and mAbs. For example, a dual-specificity TKI, lapatinib, showed synergistic inhibitory effects when combined with trastuzumab and applied on HER2-overexpressing breast cancer cell lines (Xia et al, 2002). In addition to combinations of erlotinib or afatinib and cetuximab, which were tested in clinical trials (Janjigian et al, 2011; Janjigian et al, 2014), a fourth-generation EGFR inhibitor showed synergy when combined with cetuximab in mouse models (Jia et al, 2016).

While attempting to resolve the mechanism of action of the drug triplet, we inferred two potential mechanisms of cooperation:

(i) The combination of two mAbs and a first-line TKI depletes an oncogenic transcription factor, FOXM1, and several of its target genes (e.g., AURKA, BIRC5, CENPA, cyclinB1, PRC1, and KIF4A), thus suppressing key regulators of two critical steps of the cell cycle, mitosis, and cytokinesis. Interestingly, both AURKA (Shah et al, 2018) and FOXM1 (Wang et al, 2016) have previously been implicated in resistance to EGFR TKIs. Moreover, FOXM1 can bind directly to the promoter of MET, which establishes, together with AKT, a positive feedback loop relevant not only to NSCLC (Wang et al, 2016) but also to carcinomas of tongue (Yang et al, 2018) and pancreas (Cui et al, 2016).

(ii) TKI + 2XmAbs can downregulate several RTKs previously implicated in resistance to EGFR inhibitors. The list includes, in addition to EGFR and HER2, the hepatocyte growth factor receptor, MET, a major driver of drug resistance (Engelman et al, 2007), AXL (Zhang et al, 2012), and the receptor for IGF1, which associates with resistance to osimertinib (Hayakawa et al, 2019). While mechanisms permitting co-endocytosis of several RTKs remain unknown, we assume that antibody bivalence, receptor heterodimerization, and components of the non-clathrin dependent endocytosis route mediate concurrent internalization and degradation (Mosesson et al, 2008).

In conclusion, although several EGFR TKIs are available in clinical practice, no viable treatment options are available post-emergence of resistance to third-generation TKIs like osimertinib. The results of our study offer a potentially durable treatment alternative. This would entail re-purposing of clinically approved drugs like cetuximab and trastuzumab. If applied upfront on TKI-naïve patients, the TKI + 2XmAbs triplet is expected to inhibit all mutant forms of EGFR. Further, the high efficacy of the three triplets we tested in animals implies that TKIs of all three generations might be suitable. However, continuous treatment would be needed, or tumors will relapse. Thus, the strong TKI-mAb cooperation we observed in animal models seems to warrant clinical tests for the ability to replace the current mainstay of sequential treatments making use of next-generation TKIs.

# Materials and Methods

### Cell cultures and reagents

Human NSCLC cells were obtained from ATCC (PC9) and from NIH/NCI (H3255, Bethesda, MD; (Fujishita et al, 2003)), and maintained in RPMI1640 containing 10% FBS and antibiotics. All cell lines were tested for mycoplasma. Erlotinib was from LC Laboratories, afatinib and imatinib from MedChem Express, and osimertinib was a gift from AstraZeneca. All TKIs were dissolved in DMSO at a stock concentration of 10 mM. Cetuximab and trastuzumab were obtained from Merck and Roche, respectively. The antibodies used for immunoblotting were purchased from Cell Signaling (dilution 1:1000): anti-EGFR (#4267S), pEGFR Y1068 (#2234S), HER2 (#4290S), pHER2 Y1221/1222 (#2249S), HER3 (#12708S), pHER3 (Y1289; #4791S), IGF1Rβ (#9750S), VEGFR2 (#2479S), AXL (#8661S), MET (#8198S), ERK1/2 (#4695S), pERK1/2 (#9101S), AKT1 (#2938S), pAKT (S473; #4060S), caspase3 (#9622S), cleaved caspase3 (#9661S), BIM (#2933S), Survivin (#2803S), γH2AX (#2577S), AURKA (#4718S), FOXM1 (#5436S), pFOXM1 T600 (#14655S), TPX2 (#12245S), and Cyclin B1 (#12231S). An anti-GAPDH (#MAB374) antibody was obtained from Millipore (dilution 1:15000), and the anti-α-tubulin (#PA5-58711) was obtained from Thermo Scientific (dilution 1:1000). PRC1 (#OAAN02101) and KIF4A (#OAGA05174) antibodies were obtained from Aviva Systems Biology. The FOXO3A (#2497S) antibody used for immunofluorescence was purchased from Cell Signaling (dilution 1:200). The fluorescent antibodies used for FACS were obtained from BioLegend (5 µl per million cells in 100 µl staining volume): anti-EGFR-Alexa Fluor 488 (#352908), HER2-APC (#324408), and HER3-PE (#324706).

### Cell cycle analysis

Cells were seeded in 10-cm dishes and later treated for 48 h with the indicated drugs in media containing 1% serum. Cell cycle analyses were performed using the Maxpar® Cell Cycle Panel Kit (Fluidigm). Briefly, following incubation with drugs, cells were exposed to IdU for 2 h and then incubated with metal-conjugated specific primary antibodies included in the Maxpar Cell Cycle Panel Kit. Samples were assayed using the CyTOF-2 mass cytometer.

### Determination of receptor abundance on the cell surface

Cells were seeded in 6-well plates ($1.0 \times 10^6$ per well). On the next day, media were replaced with media containing 1% serum and the cells were treated with drugs for additional 24 h. Thereafter, cells were washed in acidic buffer (glycine-HCl 100 mM, pH 3.0) and incubated with fluorescent antibodies. Surface signals were analyzed using the BD FACS Aria Fusion instrument.

### Immunofluorescence analysis

Cells were grown on coverslips in 12-well plates. Following 24-h long treatments, cells were washed in acidic buffer (glycine-HCl 100 mM, pH 3.0), followed by washes with saline, and fixation for 15 min at room temperature in paraformaldehyde (4%). Next, cells were washed and permeabilized for 10 min in saline containing Triton X-100 (0.2%). Blocking was carried out for 60 min using bovine serum albumin (1%), followed by an overnight incubation at 4°C with a primary antibody. Thereafter, cells were washed, stained for 60 min in dark with an Alexa Fluor 555- or Alexa Fluor 488-conjugated secondary antibody, and with DAPI. Images were captured using a Zeiss Spinning disk confocal microscope and processed using the Zeiss ZEN 3.1 software. To analyze tumor

specimens, formalin-fixed paraffin-embedded specimens underwent deparaffinization in xylene and rehydration in graded ethanol, followed by antigen retrieval using a citric acid solution (pH 9.0; 10 min in a microwave). The slides were washed thrice with saline, blocked in buffer containing 20% normal horse serum, and treated with an avidin/biotin blocking solution (15 min). Thereafter, sections were incubated overnight at room temperature with the corresponding primary antibodies. The slides were washed and incubated with a biotinylated secondary antibody, for 90 min at room temperature, followed by Cy3-conjugated streptavidin. After three washes, sections were incubated with DAPI to stain nuclei. Each slide was treated with mounting medium and examined on the next day using a fluorescence microscope. Positive cells were counted using Fiji.

### RNA isolation, real-time PCR, and RNA-seq analyses

Total RNA was extracted using the miRNeasy Mini Kit (QIAGEN). Complementary cDNA was synthetized using the qScript cDNA Synthesis Kit (Quantabio). Real-time quantitative PCR (qPCR) analyses were performed using SYBR green (Applied Biosystem), along with specific primers (see complete list of primers in Appendix Table S1) designed using PrimerBank on the StepOne Plus Real-Time PCR system (Applied Biosystems). qPCR signals (Ct) were normalized to GAPDH. Libraries for RNA-Seq were prepared following the bulk MARS-Seq protocol (Jaitin *et al*, 2014) and sequenced using the NextSeq500 system. Quality checks, preprocessing, alignment, and differential expression analysis were performed using the "User-friendly Transcriptome Analysis Pipeline" (UTAP) (Kohen *et al*, 2019). Differential expression analysis was performed using DESeq2 (Love *et al*, 2014). Genes were considered to be differentially expressed if their *P*-value was smaller or equal to 5e-06 and Log Fold Change threshold ± 1. The tool "Enrichr" (Chen *et al*, 2013; Kuleshov *et al*, 2016) was used to perform pathway enrichment analysis. A graph with all 15 most significantly enriched pathways from the "NCATS BioPlanet" integrative platform (Huang *et al*, 2019) was prepared. All plots and graphs related to RNA-Seq data analysis were prepared using R version 3.6.2.

### Cell viability assays

Cell viability was assessed by using MTT (3-(4,5-dimethylthiazol-2-yl)-2,5-diphenyltetrazolium bromide). PC9 cells ($5 \times 10^3$/well) or H3255 cells ($3 \times 10^4$/well) were seeded in 96-well plates. On the next day, cells were treated for 72 h with the indicated drugs. Afterward, cells were incubated for 3 h at 37°C with the MTT solution (0.5 mg/ml). The formazan crystals formed by metabolically active cells were dissolved in DMSO and the absorbance was read at 570 nm.

### Immunoblotting analyses

Protein extracts were prepared either from cell lines or from tumors extracted from mice. Tumors collected from mice were smashed using the gentleMACS™ Dissociator in RIPA buffer (50 mM Tris pH 7.5, 150 mM NaCl, 1% NP40, 0.5% sodium deoxycholate, 1 mM EDTA, 1 mM $Na_3VO_4$, and a protease inhibitor cocktail). Cells were

**The paper explained**

**Problem**
EGFR-driven lung tumors initially respond very well to drugs specific to EGFR (called: tyrosine kinase inhibitors; TKIs). However, within approximately one year, most patients acquire resistance to the drugs. Unfortunately, this cycle recurs with all next-generation TKIs, such that patients eventually succumb to their disease.

**Results**
In an attempt to establish pharmacological scenarios potentially able to replace sequential treatments with the first-, second-, and third-generation TKIs, we simulated in animals harboring human tumors the situation of TKI-naïve patients receiving first-line drugs. To prevent emergence of resistance, we admixed individual TKIs from all three generations and two clinically approved antibodies, to EGFR and to its kin, HER2. The results clearly showed that the combination of a TKI, either erlotinib or osimertinib, and the pair of antibodies prevented emergence of resistance and preempted sequential treatments with next-generation TKIs. We further report that the efficacies of upfront treatments are due to blocking biochemical pathways essential for specific steps of the cell cycle, such as mitosis and cytokinesis.

**Impact**
Patients with EGFR[+] lung cancer, who progress after sequential treatments with next-generation EGFR TKIs, have no viable treatment options, but our observations in mice offer a potential alternative. The upfront strategy proposed herein employs a combination of three drugs, all are already approved for clinical application.

washed in PBS and then extracted in RIPA buffer. Proteins were separated using gel electrophoresis and transferred to nitrocellulose membranes. After blocking, membranes were incubated overnight with the indicated primary antibodies, followed by incubation with horseradish peroxidase-conjugated secondary antibodies (1 h), and treatment with Clarity™ Western ECL Blotting Substrates (Bio-Rad). ECL signals were detected using the ChemiDoc™ Imaging System (Bio-Rad) and images were acquired using the ImageLab Software.

### Apoptosis assays

Cells were seeded in 10-cm dishes. On the next day, complete media were replaced with media containing fetal bovine serum (1%) and cells were treated for 48 h with the indicated drugs. Apoptosis was assessed using flow cytometry and the FITC Annexin V Apoptosis Detection Kit with 7-AAD (from BioLegend). The analysis was performed on the BD LSR II cytometer (BD Biosciences).

### Animal experiments

CD1 nude and NSG mice were purchased from Envigo and The Jackson Laboratory, respectively. All animal studies were approved by our institutional board, and they were performed in accordance with the guidelines of the Institutional Animal Care and Use Committee (IACUC). Mice were housed and handled in a pathogen-free, temperature-controlled (22°C ± 1°C) mouse facility on a reverse 12/12 h light/dark cycle, with lights switched on at 10 p.m. Animals were fed a regular chow diet (2018 Teklad Global 18% Protein Rodent Diet). Animals were given ad libitum access to food and

water. PC9 cells ($3 \times 10^6$ per mouse) were subcutaneously injected in the right flanks of 6-week-old female CD1 nude mice. Once tumors reached a volume of approximately 500 mm$^3$, mice were randomized into different groups and treated as indicated. TKIs were daily administered using oral gavage. Antibodies were administered twice a week using intraperitoneal injection at a final dose of 200 μg/mouse/injection. The TM00193 (exon 19del) and TM00199 (L858R) PDX models were purchased from The Jackson Laboratory and implanted in 6-week-old female and male NSG mice. Following euthanasia, tumors were removed from donor mice and cut into small fragments. A small pouch was made in the lower back of the mouse, and one fragment was later inserted into the pouch. The wound was closed using a surgical clip. Clips were removed 4–5 days after surgery. Mice were labeled with RF identification chip (from Troven). Tumors were measured twice a week with a caliper, and body weight was measured once a week. Tumor volume was calculated by using the formula $3.14 \times$ shortest diameter $\times$ (longest diameter)$^2 \times 1/6$. Mice were euthanized when tumors reached 1,500 mm$^3$ (PC9 xenografts) or 1,000 mm$^3$ (PDX models).

## Statistical analysis

Microsoft Excel, GraphPad Prism (version 8.0.2), and R (version 3.6.2) software packages were used to analyze the data. Sample numbers and other information (mean ± SEM or SD, number of replicates and specific statistical tests) are indicated in the main text or figure legends, as well as Appendix Table S2.

# Data availability

The RNA-seq datasets generated in this study are available at Gene Expression Omnibus (GEO) under the accession number GSE161584 (https://www.ncbi.nlm.nih.gov/geo/query/acc.cgi?acc=GSE161584).

**Expanded View** for this article is available online.

## Acknowledgments

We thank all members of our laboratory for their kind help and insightful comments. IM and DR would like to thank the Lombroso Foundation for their fellowships. This work was performed in the Marvin Tanner Laboratory for Research on Cancer. YY is the incumbent of the Harold and Zelda Goldenberg Professorial Chair in Molecular Cell Biology. Our studies are supported by the European Research Council (ERC) and the Dr. Miriam and Sheldon G. Adelson Medical Research Foundation.

## Author contributions

Experiment design and manuscript writing: IM and YY; Experiments: IM, DR, IV, DD-G, RO, MLU, NBN, SG, RE, TMS, and ML. Manuscript review: DR.

## Conflict of interest

The authors declare that they have no conflict of interest.

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
