## [Review Process File · EMBO Molecular Medicine]

Upfront admixing antibodies and EGFR inhibitors preempts sequential treatments in lung cancer models

Ilaria Marrocco, Donatella Romaniello, Itay Vaknin, Diana Drago, Roni Oren, Mary Luz Uribe, Nishanth Nataraj, Soma Ghosh, Raya Eilam, Tomer Salame, Moshit Lindzen, and Yosef Yarden

DOI: [10.15252/emmm.202013144](https://doi.org/10.15252/emmm.202013144)

Corresponding author: Yosef Yarden (yosef.yarden@weizmann.ac.il)

Review Timeline:

Submission Date:	20th Jul 20
Editorial Decision:	26th Aug 20
Editorial Correspondence:	1st Sep 20
Revision Received:	25th Nov 20
Editorial Decision: Revision	11th Jan 21
Received: Accepted:	29th Jan 21
	2nd Feb 21

Editor: Lise Roth

Transaction Report:

26th Aug 2020

Dear Prof. Yarden,

Thank you for the submission of your manuscript to EMBO Molecular Medicine, and please accept my apologies for the delay in getting back to you. We have now received feedback from two of the three reviewers who agreed to evaluate your manuscript. Given that referee #3 has unfortunately not returned his/her report so far despite several chasers, and that both referees 1 and 2 are overall positive, we prefer to make a decision now in order to avoid further delay in the process. Should referee #3 provide a report, we will send it to you, with the understanding that we would not ask you for extensive experiments in addition to the ones required in the enclosed reports.

As you will see from the reports below, the referees acknowledge the interest of the study and are overall supporting publication of your work pending appropriate revisions. Addressing the reviewers' concerns in full will be necessary for further considering the manuscript in our journal, with the exception of the additional PDX models mentioned by referee #1 ("Other potential PDX models from patients inhibiting acquired resistance may have been helpful."). If you do have data at hand, we will be happy for you to include them, however, it will not be mandatory for further consideration of your work. Acceptance of the manuscript will entail a second round of review. EMBO Molecular Medicine encourages a single round of revision only and therefore, acceptance or rejection of the manuscript will depend on the completeness of your responses included in the next, final version of the manuscript. For this reason, and to save you from any frustrations in the end, I would strongly advise against returning an incomplete revision.

When submitting your revised manuscript, please carefully review the instructions that follow below. Failure to include requested items will delay the evaluation of your revision:

- 1) A .docx formatted version of the manuscript text (including legends for main figures, EV figures and tables). Please make sure that the changes are highlighted to be clearly visible.
- 2) Individual production quality figure files as .eps, .tif, .jpg (one file per figure).
- 3) A .docx formatted letter INCLUDING the reviewers' reports and your detailed point-by-point responses to their comments. As part of the EMBO Press transparent editorial process, the point-by-point response is part of the Review Process File (RPF), which will be published alongside your paper.
- 4) A complete author checklist, which you can download from our author guidelines (<https://www.embopress.org/page/journal/17574684/authorguide#submissionofrevisions>). Please insert information in the checklist that is also reflected in the manuscript. The completed author checklist will also be part of the RPF.
- 5) Before submitting your revision, primary datasets produced in this study need to be deposited in an appropriate public database (see <https://www.embopress.org/page/journal/17574684/authorguide#dataavailability>).

Please remember to provide a reviewer password if the datasets are not yet public. The accession numbers and database should be listed in a formal "Data Availability " section (placed after Materials & Method). Please note that the Data Availability Section is restricted to new primary data that are part of this study.

6) We would also encourage you to include the source data for figure panels that show essential data. Numerical data should be provided as individual .xls or .csv files (including a tab describing the data). For blots or microscopy, uncropped images should be submitted (using a zip archive if multiple images need to be supplied for one panel). Additional information on source data and instruction on how to label the files are available at .

7) Our journal encourages inclusion of *data citations in the reference list* to directly cite datasets that were re-used and obtained from public databases. Data citations in the article text are distinct from normal bibliographical citations and should directly link to the database records from which the data can be accessed. In the main text, data citations are formatted as follows: "Data ref: Smith et al, 2001" or "Data ref: NCBI Sequence Read Archive PRJNA342805, 2017". In the Reference list, data citations must be labeled with "[DATASET]". A data reference must provide the database name, accession number/identifiers and a resolvable link to the landing page from which the data can be accessed at the end of the reference. Further instructions are available at .

8) We replaced Supplementary Information with Expanded View (EV) Figures and Tables that are collapsible/expandable online. A maximum of 5 EV Figures can be typeset. EV Figures should be cited as 'Figure EV1, Figure EV2" etc... in the text and their respective legends should be included in the main text after the legends of regular figures.

- Additional Tables/Datasets should be labeled and referred to as Table EV1, Dataset EV1, etc. Legends have to be provided in a separate tab in case of .xls files. Alternatively, the legend can be supplied as a separate text file (README) and zipped together with the Table/Dataset file. See detailed instructions here: .

9) For more information: There is space at the end of each article to list relevant web links for further consultation by our readers. Could you identify some relevant ones and provide such information as well? Some examples are patient associations, relevant databases, OMIM/proteins/genes links, author's websites, etc...

10) Every published paper now includes a 'Synopsis' to further enhance discoverability. Synopses are displayed on the journal webpage and are freely accessible to all readers. They include a short stand first (maximum of 300 characters, including space) as well as 2-5 one-sentences bullet points that summarizes the paper. Please write the bullet points to summarize the key NEW findings. They should be designed to be complementary to the abstract - i.e. not repeat the same text. We encourage inclusion of key acronyms and quantitative information (maximum of 30 words / bullet

point). Please use the passive voice. Please attach these in a separate file or send them by email, we will incorporate them accordingly.

Please also suggest a striking image or visual abstract to illustrate your article. If you do please provide a png file 550 px-wide x 400-px high.

11) As part of the EMBO Publications transparent editorial process initiative (see our Editorial at <http://embomolmed.embopress.org/content/2/9/329>), EMBO Molecular Medicine will publish online a Review Process File (RPF) to accompany accepted manuscripts.

In the event of acceptance, this file will be published in conjunction with your paper and will include the anonymous referee reports, your point-by-point response and all pertinent correspondence relating to the manuscript. Let us know whether you agree with the publication of the RPF and as here, if you want to remove or not any figures from it prior to publication.

I look forward to receiving your revised manuscript.

Yours sincerely,

Lise Roth

Lise Roth, PhD
Editor
EMBO Molecular Medicine

To submit your manuscript , please follow this link:

Link Not Available

Each figure should be given in a separate file and should have the following resolution:
Graphs 800-1,200 DPI
Photos 400-800 DPI
Colour (only CMYK) 300-400 DPI"

*Additional important information regarding figures and illustrations can be found at <http://bit.ly/EMBOPressFigurePreparationGuideline>

***** Reviewer's comments *****

Referee #1 (Comments on Novelty/Model System for Author):

In the preclinical cell line models only 2 cell lines were used one with exon 19 and the other with exon 21 alterations. Ideally more cell lines would have been better.

Other potential PDX models from patients inhibiting acquired resistance may have been helpful. What can help here is the combination helpful in treating acquired resistance in addition to preventing resistance.

Referee #1 (Remarks for Author):

This is a study looking at the combination of EGFR TKI's and monoclonal antibodies targeting EGFR and HER2 in cell lines and animal models (human derived tumors) to determine if the combos are more effective than single agent therapy. They also look at upstream receptors, downstream signaling and changes in gene expression.

- 1- The main criticism of this manuscript relates to the timing of it. Studies of combination therapies in EGFR mutant lung cancer are now over 10 years old and no benefit to the addition of cetuximab has been seen.
- 2- Furthermore if blocking HER2 would have added to progression free survival in these patients one would postulate that afatinib that blocks EGFR and HER2 would have done better in clinical trials.
- 3- the field of enhancing EGFR targeting has moved away from HER2 and EGFR combination therapy

Referee #2 (Comments on Novelty/Model System for Author):

The work is of generally high quality, but completely lacks statistical analyses - even for the data where such an analysis is quite trivial. Given that the work builds on the previous publications from the lab, the novelty is moderate. However, the study has potentially highly important clinical implications, thus clinical impact is high.

Referee #2 (Remarks for Author):

The manuscript by Marrocco et builds on this group's prior work (Mancini et al., 2015) and demonstrates that addition of EGFR and HER2 antibodies to EGFR TKI therapy dramatically

increases tumor responses in vivo (in PC9 and 2 xenograft models), and can even lead to cures in animal models. Moreover, this combination therapy can be effective in tumors that relapse on EGFR TKI monotherapies. Importantly, the enhanced efficiency does not appear to come at the expense of increased systemic toxicity (at least based on animal weight metrics). Therefore, the translational potential of these findings is high. The paper also aims to dissect the underlying mechanisms, concluding that FOXM1 dependent mitotic regulation is responsible for the observed additive effect. The mechanistic part of the paper is substantially weaker, as authors appear to over-interpret their observations, and, in many cases, the stated results are not quite supported by the data. In addition, authors completely avoid statistical analysis of the data, even in cases where such analyses are clearly called for.

Specifically:

1. Mechanistic part of the study is purely observational, yet authors draw strong conclusions. If authors are correct, and the reason for the enhanced efficiency of the combination therapy is indeed the FOXM1 dependent defective mitosis and cytokinesis, one would expect to detect a significant fraction of G2 arrested tumor cells - which can be easily tested. However, this mechanism is highly unlikely, as TKIs induce strong G1/G0 cell cycle arrest (which authors actually show in EV3). Authors should either perform a rigorous interrogation of underlying mechanism, or tone down their conclusions to reflect the level of evidence that their data shows.
2. Suppression of non-targeted RTKs by the combination of the TKIs with antibody is remarkable and is likely responsible for the observed synergy. I am surprised at not going beyond the speculations on the putative mechanisms here - especially considering that the main putative mechanism above is entirely based on a conjecture.
3. Authors avoid statistical analysis of the data, and in many instances the stated results are inconsistent with the data. Most of these can be easily fixed by using less categorical language, but it needs to be addressed. Some examples are provided below:
 - "Notably, combining 2XmAbs and each of the three TKIs consistently increased these effects." Data presented in EV3B does not have any statistical analyses. Moreover, the eyeballing shows the increase only with erlotinib, but not with osimertinib or afatinib. Thus, "These effects were reproducible and consistent, since they were shared by two different cellular models and three TKIs" does not reflect the actual data.
 - Data shown in Figure 1b should be analyzed for statistical significance
 - It will be very helpful to quantify the western blotting data (1a, 4b, 7c, plus supplementary), even in cases when single replicate was used, as this will make comparative statements easier to evaluate.
 - "HER3 displayed an opposite response: all drugs strongly elevated HER3 abundance, but almost completely erased HER3 phosphorylation" HER3 data from Figure 1B is not necessarily consistent with the data shown in 1A. The extent of HER3 increase is substantially lower in whole cell lysates, and for afatinib combination, HER3 levels appear to be lower than in the control in the PC9 cells.
4. Instead of relying on the use of WB mediated analysis of apoptotic markers, use of cell-based analysis (such as IF/IHC or flow cytometry) would have been much more informative, for both in vitro and in vivo experiments.
5. "Notably, due to relatively high abundance of BIM and cleaved caspase 3 in the control mice, we were unable to detect alterations in these markers of apoptosis." For the in vivo analyses, it would have been highly informative to analyze proliferation status of tumor cells. High levels of apoptosis mediated cell turnover within tumors, even in the absence of treatment, is not surprising - but it makes it more challenging to interpret the data. Since growth and regression of tumors reflect net balance between proliferation and death, it would have been highly useful to complement the analyses shown in Figure 4 by assessing proliferation levels within tumors. Ideally, proliferation and apoptosis data should be assessed by histological analyses (IF or IHC) - if authors have samples of fixed tumor tissue from these experiments. I think that the interpretation of the in vivo data after 7 days of treatment is more complicated than a short-term in vitro assay - as one deals with cells

that avoided initial elimination, and cell cycle arrest induced by TKIs might reduce the proportion of cells where apoptosis can be induced by the antibodies - thus even decrease in % of apoptotic cells in the combination group compared to the Ab one might be still compatible with enhanced cell death rates in the combination treatment.

6 . For the RNA seq data presented in Figure 7 and Supplement, insufficient technical detail is provided. What are the criteria for the differential expression? Interestingly, there is no obvious difference between Erlotinib responding and non-responding tumors, in expression of genes differentially expressed between the controls and the combination group. Why only downregulated genes are analyzed for the Pathway analysis? By itself, the expression data does not enable to make strong conclusions that the authors make in terms of the underlying mechanisms. While functional validation would be ideal, at the very least the authors can dig deeper into the expression data that they already have.

Minor comments

1. "No previous studies examined emergence of resistance to mAbs, or mAbs+TKI combinations" - this is clearly wrong (at least as stated, perhaps authors had something else in mind - in which case a more careful wording is warranted). There is plenty of literature on mechanisms of resistance to trastuzumab and other antibody-based therapies.
2. Results, first paragraph, page 4, line 7-8, "EGFR-specific TKIs" is unnecessarily repeated in the sentence.
3. Authors use different concentrations of TKIs in different in vivo experiments (such as 5 and 10 mg/kg osi), without providing a rationale for these differences.
4. It would have been highly helpful to show IdU-PI plots to characterize the impact of different treatments on cell cycle progression (in EV3). This would be more informative than using IdU by itself.
5. Consider being consistent when referring to the Expanded view data, as both Expanded view of figures (EV) and supplementary figures (S) notations are used in the main text. e.g, Page 6, second paragraph line 7, Figure S4--EV4; Page 10, line3, S7 -EV7; Page12, line 9, S8A-EV8A;
6. Consider using consistent notation of TKI+2XmAbs throughout the paper when the combination treatment was conducted.
7. EV1 legend refers to the non-existent panel E, instead of the C.
8. Consider indicating the cell line or xenografts name on western blot images throughout the results figures sections as authors did on Fig1A.
9. "all combos associated with downregulation of IGF1R" - "combo" is overly colloquial.

1st Sep 20

Dear Prof. Yarden,

We just received the review from referee #3 that you will find below. As you will see, this referee mostly requires a deeper mechanistic understanding, which is in line with the previous reports. As mentioned in my decision letter, we will not ask you for further reaching experiments at this point, but would like you to address as much as possible the comments 1, 4 and 5 from this referee.

Looking forward to receiving your revised manuscript,

With my best wishes,

Lise Roth

Referee #3:

This is an impressive paper representing a very complete study of EGFR inhibitor resistance in mouse models. There are a lot of animal expts to unpack here I commend the authors on the rigor of the study. The major mechanistic insight is that the combination of TKI+2xMABS cause a more pronounced inhibition of pERK and to some extent pAKT in vivo, causing extended anti-tumor control. This effect is mediated by proposed blocking of feedback loops that cause re-activation of this pathway when using EGFR TKIs alone. This is a potentially suitable paper provided some more in depth analysis can be done to elucidate the mechanistic basis for the triple drug combo.

1. Perhaps the most pressing issue raised by this manuscript is a need for a more complete molecular characterization of mechanisms of resistance to EGFR TKIs in vivo (Fig 5, 6C). Given the efficacy of the triple combo, it appears the basic mechanism of resistance assumed to be at play in these settings is reactivation of EGFR an/or HER2. This needs to be demonstrated by analysis of relapsed tumor tissue for these phospho-proteins and downstream pathways. Without this knowledge one wonders whether the effects observed are truly on-target

2. Fig 5 -In vivo sequential treatments suggest that resistance to TKIs is mediated by EGFR/HER2 and respond to 2xMabs. This is shown in the PC9 model where such activity may be explained by the emergence of t790M in the case of Erlotinib, or something similar in the case of Osimertinib (perhaps C797S). Can the authors do sequence analysis of the in vivo resistant tumors to identify potential drivers of resistance in this model?

3. The in vivo sequential study is highly dependent on the nature of drug resistance mechanism that may be different between models. This is critical to understand the breadth of tumor control expected with this combination, one could test this by applying a similar approach to Fig 5 with another model

4. Fig 6- analysis of relapsed tumors indicating that they remain dependent on EGFR/HER2

in panel C. These studies are not analogous to residual disease in patients or in cell lines (eg Hata et al, Ramirez et al) as they are under continuous drug treatment which is not true in this case. Clearly the residual disease state is not dependent on EGFR/HER2 because the cells that repopulate the tumor survive the initial treatment with Er+2xmABS. Hence it seems that drug holiday after residual disease leads to re-emergence of dependence on EGFR/HER2, but this dependency is not present at the RD stage itself. Analysis of signaling through serial sampling (upfront, RD, relapse) can be used to determine when EGFR/HER2 are most active.

5. Fig 7, what happens to HER3 in this model? Consistent with RTK reactivation due to loss of negative feedback RTKs should be upregulated - please confirm. Also, does the Er-resitant tumor show reactivation of EGFR and/or activation of HER2? This is critical for explaining the mechanism of 2XmAbs.

EMBO Molecular Medicine Manuscript: EMM-2020-13144

Ilaria Marrocco et al., *Upfront admixing antibodies and EGFR inhibitors preempts sequential treatments in lung cancer models*

Referee #1 (Comments on Novelty/Model System for Author):

In the preclinical cell line models only 2 cell lines were used one with exon 19 and the other with exon 21 alterations. Ideally more cell lines would have been better.

Our study employed two cell lines, each harboring a distinct highly prevalent EGFR mutation: PC9 cells, harboring an exon 19 deletion, and H3255 cells carrying the L858R point mutation. In response to this comment, we purchased two additional NSCLC cell lines. These are HCC4006, harboring the L747 - E749 deletion (A750P), and HCC2935 cells, carrying the E746 - T751 deletion (S752I), and performed the experiments presented below (Fig. I). Unfortunately, unlike the more commonly used PC9 and H3255 cell lines, both new lines underwent only very slow proliferation in vitro, which prevented in-depth analyses of their responses and many experimental replicates, as done with the other two cell lines. Nevertheless, the results we obtained supported, in general, the conclusions reached on the basis of the other cell lines. For example, proliferation assays presented in panel A confirmed that the three TKIs more strongly than 2XmAbs inhibited cell proliferation, and the addition of 2XmAbs augmented the inhibitory effects. In addition, analyses of apoptosis markers (Fig. I-B) were in line with the results we obtained when using PC9 and H3255 cells. For example, the addition of antibodies, in general, enhanced TKI-induced downregulation of survivin, along with up-regulation of BIM and cleavage of caspase 3. In addition, TKIs enhanced expression of EGFR in HCC4006 cells, whereas treatments that made use of a TKI plus 2XmAbs commonly downregulated EGFR (Fig. I-C). As expected, HER2 and MET underwent downregulation in response to treatment of both HCC2935 and HCC4006 with combinations of TKIs and antibodies. Note that HER3 was upregulated by all TKI+2XmAbs, save for afatinib+2XmAbs.

In summary, the new results we obtained while using two additional cell lines harboring exon 19 deletions supported the observations we made with PC9 and H3255 cells. However, due to space considerations, as well as due to the difficulties we faced with the slow doubling times of the new cell lines, we could not include the new data in the revised manuscript.

Figure I: When combined with EGFR-specific TKIs, monoclonal anti-EGFR and anti-HER2 antibodies reduce viability and enhance apoptosis of previously untested cell lines harboring exon 19 deletions. (A) HCC2935 (E746

- T751 deletion, S752I; 9X10⁴) and HCC4006 cells (L747 - E749 deletion, A750P; 9X10⁴) were seeded in 96-well plates and treated for 72 hours with increasing concentrations of TKIs (erlotinib or osimertinib, each at 10 and 100 nM) either alone or in combination with 2XmAbs (cetuximab plus trastuzumab, each at 5 µg/ml). Cell viability was assessed using the MTT assay. Data, obtained from 5 replicates are shown as mean + SEM. (B) HCC2935 and HCC4006 cells were treated for 48 hours with increasing concentrations of EGFR-specific TKIs (erlotinib, osimertinib or afatinib at 5, 10, 20, 40 and 80 nM), either alone or in combination with 2XmAbs (cetuximab plus trastuzumab, each at 5 µg/ml). Protein extracts were resolved using electrophoresis, transferred to nitrocellulose membranes and blotted with the indicated antibodies. GAPDH was used as a loading control. Signals (relative to control) were quantified and normalized to the signals of GAPDH. (C) HCC2935 and HCC4006 cells were treated for 24 hours with 2XmAbs (cetuximab plus trastuzumab, each at 5 µg/ml), either alone or in combination with the following TKIs: erlotinib or osimertinib (each at 50 nM), or afatinib (10 nM). Protein extracts were immunoblotted using the indicated antibodies. Signals (relative to control) were quantified and normalized to the signals of GAPDH or tubulin.

Other potential PDX models from patients inhibiting acquired resistance may have been helpful. What can help here is the combination helpful in treating acquired resistance in addition to preventing resistance.

1. Two PDX models TM00193 (exon 19 deletion) and TM00199 (exon 21 point mutation) have been used in our study, along with the PC9 cell line xenograft. Purchasing a third xenograft model from the Jackson Laboratory has been initiated, but shipping and establishing the new model in our PDX Model Unit might take several additional months.
2. Figure 5 of the revised manuscript shows animal studies performed with mice that acquired resistance to either erlotinib (panel A) or osimertinib (panel B). As demonstrated, adding 2XmAbs on top of the TKI, can be effective in tumors that relapsed on EGFR TKI monotherapy. Thus, all mice we tested were practically cured.

Referee #1 (Remarks for Author):

This is a study looking at the combination of EGFR TKI's and monoclonal antibodies targeting EGFR and HER2 in cell lines and animal models (human derived tumors) to determine if the combos are more effective than single agent therapy. They also look at upstream receptors, downstream signaling and changes in gene expression.

1- The main criticism of this manuscript relates to the timing of it. Studies of combination therapies in EGFR mutant lung cancer are now over 10 years old and no benefit to the addition of cetuximab has been seen.

We definitely agree that adding cetuximab has no benefit, both in patients and in animal models. The studies presented herein indicate that concurrent blockade of *both* HER2 and EGFR can confer long-term benefit.

2- Furthermore if blocking HER2 would have added to progression free survival in these patients one would postulate that afatinib that blocks EGFR and HER2 would have done better in clinical trials.

The reviewer refers to afatinib, a broad-specificity kinase inhibitor, and raises an interesting point. However, the mechanisms of action of kinase inhibitors, like afatinib or erlotinib, are clearly different from the mechanism underlying the therapeutic effects of antibodies, such a trastuzumab. For example, antibodies target their antigens to degradation and recruit effector cells of the immune system, while inducing growth arrest rather than apoptosis. However, none of these activities are shared by kinase inhibitors. As we demonstrate in Figure 3 of the revised manuscript, blocking HER2 using a TKI, such as afatinib, is less effective than blocking HER2 using a monoclonal antibody.

3- the field of enhancing EGFR targeting has moved away from HER2 and EGFR combination therapy

To try and convince the referee that simultaneously targeting both EGFR and HER2 using specific antibodies can enhance the therapeutic effects of erlotinib or the combination of erlotinib and cetuximab, we performed a new animal study that followed a first-line scenario. To this end, we implanted PC9 cells in the flanks of immunocompromised mice and, when tumors became palpable, treated the mice for three weeks only (grey areas) with the following drugs:

- erlotinib alone (50 mg/kg/day, daily)
- cetuximab alone (0.2 mg/mouse/injection, once every three days)
- cetuximab combined with erlotinib (50 mg/kg/day, daily)
- cetuximab plus trastuzumab (total antibody dose: 0.2 mg/mouse/injection)
- cetuximab plus trastuzumab (0.2 mg/mouse/injection) plus erlotinib (50 mg/kg/day, daily)

The results presented below in Figure II validated superiority of the triplet containing an anti-HER2 antibody over a doublet comprising cetuximab and erlotinib.

Figure II: The addition of an anti-HER2 antibody improves efficacy of a drug combination comprising erlotinib and cetuximab. PC9 cells (3×10^6 /mouse) were subcutaneously implanted in the flanks of CD1-nu/nu mice. When tumors became palpable, mice were randomized into groups of 5-9 animals and treated for 3 weeks (grey areas) with erlotinib alone (50 mg/kg/day, daily), cetuximab alone (0.2 mg/mouse/injection, once every three days), cetuximab combined with erlotinib (50 mg/kg/day, daily), 2XmAbs (cetuximab plus trastuzumab, 0.2 mg/mouse/injection) or with

2XmAbs combined with erlotinib. Shown are the averaged tumor volumes (A) and animal survival times (B). Mice were euthanized when tumor size reached 1,500 mm³. Data are means ± SEM from 5-9 mice per group. Tumor volumes of individual mice of each group are shown in C.

Referee #2 (Comments on Novelty/Model System for Author):

The work is of generally high quality, but completely lacks statistical analyses - even for the data where such an analysis is quite trivial. Given that the work builds on the previous publications from the lab, the novelty is moderate. However, the study has potentially highly important clinical implications, thus clinical impact is high.

We thank the referee for these comments. As detailed below, the issue of statistical analyses has been corrected in the revised manuscript.

Referee #2 (Remarks for Author):

The manuscript by Marrocco et builds on this group's prior work (Mancini et al., 2015) and demonstrates that addition of EGFR and HER2 antibodies to EGFR TKI therapy dramatically increases tumor responses in vivo (in PC9 and 2 xenograft models), and can even lead to cures in animal models. Moreover, this combination therapy can be effective in tumors that relapse on EGFR TKI monotherapies. Importantly, the enhanced efficiency does not appear to come at the expense of increased systemic toxicity (at least based on animal weight metrics). Therefore, the translational potential of these findings is high. The paper also aims to dissect the underlying mechanisms, concluding that FOXM1 dependent mitotic regulation is responsible for the observed additive effect. The mechanistic part of the paper is substantially weaker, as authors appear to over-interpret their observations, and, in many cases, the stated results are not quite supported by the data. In addition, authors completely avoid statistical analysis of the data, even in cases where such analyses are clearly called for.

Specifically:

1. Mechanistic part of the study is purely observational, yet authors draw strong conclusions. If authors are correct, and the reason for the enhanced efficiency of the combination therapy is indeed the FOXM1 dependent defective mitosis and cytokinesis, one would expect to detect a significant fraction of G2 arrested tumor cells - which can be easily tested. However, this mechanism is highly unlikely, as TKIs induce strong G1/G0 cell cycle arrest (which authors actually show in EV3). Authors should either perform a rigorous interrogation of underlying mechanism, or tone down their conclusions to reflect the level of evidence that their data shows.

As requested, we softened all conclusions in Results and Discussion, which are relevant to the interpretation of the cell cycle mechanisms underlying efficacy of the combination therapy.

2. Suppression of non-targeted RTKs by the combination of the TKIs with antibody is remarkable and is likely responsible for the observed synergy. I am surprised at not going beyond the speculations on the putative mechanisms here - especially considering that the main putative mechanism above is entirely based on a conjecture.

We share the understanding that the unknown mechanism enabling suppression of non-targeted receptors might provide a key to understanding the observed pharmacological synergy. Unfortunately, the data we collected so far are yet unable to provide a compelling model. Hence, the revised text was modified in line with this comment. The new text reads as follows:

“A clear reduction in the abundance of the receptors for the hepatocyte growth factor (MET), GAS6 (AXL) and the insulin-like growth factor 1 (IGF1R) was observed when either cell line was exposed to TKI+2XmAbs. Although the mechanisms underlying trans-downregulation of other

RTKs remain unclear, they could explain drug interactions. For instance, suppression of non-targeted RTKs may be mediated by heterodimer formation between EGFR and MET, IGF1R or AXL (Balana et al., 2001; Brand et al., 2017; Jo et al., 2000)."

3. Authors avoid statistical analysis of the data, and in many instances the stated results are inconsistent with the data. Most of these can be easily fixed by using less categorical language, but it needs to be addressed.

- (i) As requested, we toned down the conclusions of the corresponding parts of the revised manuscript.
- (ii) We requested advise from Dr. Ron Rotkopf, Head of our Statistics Consulting Unit. Following his recommendations, we added statistical analysis of tumor volumes corresponding to the last measurement performed with each mouse (using one-way ANOVA followed by Tukey's multiple comparison test). Analyses of the significant comparisons are shown in panels C of Figures III, IV, V and VI. All 4 new figures shown below were respectively inserted in the revised manuscript, as Figures 2 and 3, as well as Expanded View Figure 3 and Appendix Figure S3.
- (iii) We incorporated in the Appendix of the revised manuscript a new table listing all statistical parameters. Please find the table in the end of this letter.

Figure III (Fig. 2 of the revised manuscript): Neither erlotinib nor 2XmAbs are effective, but the combination cures a xenograft model driven by a single-mutation EGFR. PC9 cells (3×10^6 /mouse) were subcutaneously implanted in the flanks of CD1-nu/nu mice. When tumors became palpable, mice were randomized in groups of 5-9 animals and treated for 90 days (grey areas) with 2XmAbs (cetuximab plus trastuzumab, 0.2 mg/mouse/injection), once every three days, or with erlotinib (50 mg/kg/day), once per day. Alternatively, mice were treated with combinations of erlotinib (50, 20, 10, 5 or 1 mg/kg) and the two monoclonal antibodies. Following 60 days of treatment, we reduced the frequency of erlotinib administration to once every other day (underneath dotted line). Tumor volumes (**A**) and animal survival (**B**) are shown. Mice were euthanized when tumor size reached $1,500 \text{ mm}^3$. Data are means \pm SEM from 5-9 mice per group. (**C**) Statistical analysis of tumor volumes corresponding to the last measurement performed with each mouse. The analysis used one-way ANOVA followed by Tukey's multiple comparison test. Note that only the significant comparisons are shown. Non-significant comparisons are not indicated in the panel for reasons of space. All p-values corresponding to the data are listed in Appendix Table S2. (**D**) Shown are tumor volumes of individual mice of each group. Note that each animal is represented by a different color. The respective numbers of tumor-free mice are indicated.

Figure IV (Figure 3 of the revised manuscript): A combination of mAbs specific to EGFR and to HER2 collaborates with both second- and third-generation EGFR TKIs, but a BCR-ABL TKI displays no cooperative effects. PC9 cells (exon 19 deletion) were subcutaneously implanted in the flank of CD1-nu/nu mice (3×10^6 /mouse). When tumors became palpable, mice were randomized in groups of 5-9 animals and treated for 90 days (grey areas) with 2XmAbs (cetuximab plus trastuzumab, each at 0.1 mg/mouse/injection) once every three days, or daily with different TKIs: osimertinib (5 mg/kg), afatinib (5 mg/kg) or imatinib (100 mg/kg), either alone or in combination with the two antibodies. Following 60 days of treatment, the frequency of TKI administration was reduced to once every other day (underneath dotted lines), while mAb treatment remained unaltered. Tumor growth (A) and animal survival (B) are shown. Mice were euthanized when tumor size reached $1,500 \text{ mm}^3$. Data are means \pm SEM from 5-9 mice of each group. (C) Statistical analysis of tumor volumes corresponding to the last measurement for each mouse, which was performed using one-way ANOVA followed by Tukey's multiple comparison test. Note that only the significant comparisons are shown. All p-values corresponding to the data are listed in Appendix Table S2. (D) Shown are tumor

volumes corresponding to individual animals of each group. The numbers of mice with undetectable tumors are indicated.

Figure V (Figure EV3 of the revised manuscript): Short-term treatments of xenografts with combinations of monoclonal antibodies and either erlotinib or osimertinib achieve similar anti-tumor efficacies. (A and B) PC9 cells (exon 19 deletion) were subcutaneously implanted in the flanks of CD1-nu/nu mice (3×10^6 /mouse). When tumors became palpable, mice were randomized into groups of 4-10 animals, which were treated for 30 days (grey area) with 2XmAbs (cetuximab and trastuzumab, each at 0.1 mg/mouse/injection) once every three days, or daily with the indicated TKIs, osimertinib (10 mg/kg) or erlotinib (50 mg/kg). A third group was treated with the respective combinations of a TKI and the two mAbs. Tumor growth (A) and animal survival (B) are shown. Mice were euthanized when tumor size reached 1,500 mm³. Data are means \pm SEM from 4-10 mice per group. (C) Statistical analysis of tumor volumes corresponding to the last measurement for each mouse was performed using one-way ANOVA followed by Tukey's multiple comparison test. Note that only the significant comparisons are shown. Non-significant

comparisons are not indicated in the panel for reason of space. All p-values corresponding to the data are listed in Appendix Table S2. (D) Shown are tumor volumes corresponding to individual mice.

Figure VI (Appendix Figure S3 of the revised manuscript): Adding two monoclonal antibodies (cetuximab and trastuzumab) to an EGFR-specific TKI either cures or significantly delays relapses of an EGFR-mutated xenograft model of NSCLC. (A and B) PC9 cells (exon 19 deletion) were subcutaneously implanted in the flanks of CD1-nu/nu mice (3×10^6 /mouse). Once tumors became palpable, mice were randomized to 8 groups of 5-9 animals and treated, once every three days, with 2XmAbs (cetuximab and trastuzumab, each at 0.1 mg/mouse/injection), the following TKIs: erlotinib (50 mg/kg/day), afatinib (2.5 mg/kg) or osimertinib (5 mg/kg), or the respective combinations of TKIs and antibodies. Note that the dose of osimertinib in the combination group was reduced to 1 mg/kg/day. Treatments were stopped on day 54 (21 days for the afatinib group). Tumor volumes (panel A) were followed for 23 additional days, and animal survival (panel B) was followed for 126 additional days. Mice were euthanized when tumor size reached 1,500 mm³. Data are means \pm SEM from 5-9 mice per group. (C) Statistical analysis of tumor volumes corresponding to the last measurement for each mouse was performed using one-way ANOVA followed by Tukey's multiple comparison test. Note that only significant comparisons are shown. Non-significant comparisons are not

indicated for reasons of space. All p-values corresponding to the data are listed in Appendix Table S2. **(D)** Tumor volumes of individual mice in each group are shown in different colors. The numbers (N) of tumor-free mice in each group are indicated.

Some examples are provided below:

• "Notably, combining 2XmAbs and each of the three TKIs consistently increased these effects." Data presented in EV3B does not have any statistical analyses. Moreover, the eyeballing shows the increase only with erlotinib, but not with osimertinib or afatinib. Thus, "These effects were reproducible and consistent, since they were shared by two different cellular models and three TKIs" does not reflect the actual data.

We thank the referee for this comment. Repeated statistical analyses shown in Figure VII, below, confirmed that 2XmAbs reduced the M-phase fraction, as did the three TKIs we tested. However, the numbers of cells found at G0 were not significantly affected. Similarly, the number of Ki67-positive cells did not significantly change in response to drug treatments. As requested, we modified the figure (Appendix Figure S2 of the revised manuscript), added statistical analyses and changed the corresponding text.

Figure VII (Appendix Figure S2 of the revised manuscript): Combinations of EGFR-specific TKIs and two monoclonal antibodies (cetuximab and trastuzumab) induce cell cycle arrest and strongly inhibit proliferation of PC9 cells. PC9 cells were seeded in 10-cm dishes and treated for 48 hours with erlotinib (40 nM), osimertinib (40 nM) or afatinib (10 nM), either alone or in combination with 2XmAbs (cetuximab and trastuzumab, each at 5 µg/ml). Thereafter, the cells were exposed to IdU (5-Iodo-2'-deoxyuridine) for 2 hours, followed by incubation with metal-conjugated primary antibodies included in the Maxpar Cell Cycle Panel Kit (Fluidigm). Samples were analyzed using the CyTOF 2 mass cytometer (Fluidigm). The percentages of cells in G0/G1, S or G2/M phases of the cell cycle were estimated according to the levels of cyclinB1. Antibodies against pRB and phosphorylated histone H3 were used to assay the fractions of cells found in G0 and M phases, respectively. Signal quantification in each phase is presented. The fraction of cells found to be in the active proliferation state are presented as Ki67⁺ cells. Data were analyzed using the FlowJo software and the results from three experiments are presented as mean + SEM. Significance was assessed using one-way ANOVA followed by Dunnett's multiple comparisons test. All p-values corresponding to the data are listed in Appendix Table S2.

• Data shown in Figure 1b should be analyzed for statistical significance

As requested, we added statistical analysis of the data shown in Figure 1B (see Figure VIII, below).

Figure VIII (Fig. 1B of the revised manuscript): Combinations of mAbs and EGFR-specific TKIs alter abundance of several receptors for survival factors. PC9 cells (1×10^6) were seeded in 6-well plates and treated for 24 hours with different EGFR-specific TKIs (erlotinib, 50 nM; osimertinib, 50 nM, or afatinib, 10 nM), either alone or in combination with 2XmAbs (cetuximab and trastuzumab, 5 $\mu\text{g/ml}$ each). After washing with acidic buffer (glycine 100 mM, pH 3.0), cells were incubated with fluorescently-labelled antibodies against EGFR, HER2 and HER3, and surface expression levels of each receptor were analyzed using flow cytometry. The normalized fluorescence intensity is shown as averages of four experiments. Significance was assessed using two-way ANOVA followed by Dunnett's multiple comparisons test. Note that non-significant comparisons are not indicated. All p-values corresponding to the data are listed in Appendix Table S2.

• It will be very helpful to quantify the western blotting data (1a, 4b, 7c, plus supplementary), even in cases when single replicate was used, as this will make comparative statements easier to evaluate.

As requested, we quantified all western blotting data and inserted the new panels in all figures. In addition, we collated all revised panels (after quantification) and inserted them in the Appendix of this letter.

• "HER3 displayed an opposite response: all drugs strongly elevated HER3 abundance, but almost completely erased HER3 phosphorylation" HER3 data from Figure 1B is not necessarily consistent with the data shown in 1A. The extent of HER3 increase is substantially lower in whole cell lysates, and for afatinib combination, HER3 levels appear to be lower than in the control in the PC9 cells.

We agree that there is some variation in the data related to HER3. This is why we performed three different analyses (i.e., western blotting, FACS and immunofluorescence), which quantified HER3 abundance. Specifically, HER3 increased primarily in the plasma membrane, such that the changes

we detected in whole cell lysates were smaller than the effects we observed using FACS or immunofluorescence.

In response to this comment, we introduced the following changes in the revised manuscript:

- (i) The revised text better highlights the differences between Figure 1A, which is a western blot analyzing the total cellular amount of HER3, and Figure 1B, which depicts surface levels of HER3, as determined using cytometry.
- (ii) The revised text also relates to potential existence of an intracellular pool of HER3 molecules and their regulation by dephosphorylation, as previously reported (Sergina et al., 2007).
- (iii) We added a comment dealing with the ability of the immunofluorescence analysis we performed to resolve both surface and intracellular HER3 molecules, while clearly detecting HER3 up-regulation following treatment with TKIs.
- (iv) We quantified all bands in western blots shown in the manuscript. Please find the quantification of data in the appendix of this document.

4. Instead of relying on the use of WB mediated analysis of apoptotic markers, use of cell-based analysis (such as IF/IHC or flow cytometry) would have been much more informative, for both in vitro and in vivo experiments.

- (i) To answer this request, we performed immunofluorescence analysis employing an antibody against cleaved caspase 3. This analysis was performed both in vitro, using PC9 cells (Figure IX; Appendix Figures S1A and S1B of the revised manuscript) and H3255 cells (see Figure X; Appendix Figures S1C and S1D of the revised manuscript), and in tumor sections (see Figure XI; Appendix Figures 4A and 4B of the revised manuscript).
- (ii) Another cell-based analysis of apoptosis we used was flow cytometry. The results of assays using annexin V in combination with 7-AAD are shown in Figure EV1-C of the revised manuscript.

Figure IX (Appendix Figures S1A and S1B of the revised manuscript): Combinations of antibodies and TKIs increase apoptosis of PC9 NSCLC cells. PC9 cells were seeded on coverslips and treated for 48 hours with different EGFR-specific TKIs (erlotinib, 40 nM; osimertinib, 40 nM, or afatinib, 10 nM), either alone or in combination with 2XmAbs (cetuximab and trastuzumab, 5 μ g/ml each). Cells were fixed in paraformaldehyde (4%) and incubated with an anti-cleaved caspase 3 antibody (CC3), followed by an Alexa Fluor 488-conjugated secondary antibody. DAPI (blue) was used to stain nuclei. Images were captured using a confocal microscope (20X magnification). Bar, 20 μ m. Significance was assessed using one-way ANOVA followed by Tukey's multiple comparisons test. Note that non-significant comparisons are not indicated. All p-values corresponding to the data are listed in Appendix Table S2.

Figure X (Appendix Figures S1C and S1D of the revised manuscript): Combinations of antibodies and TKIs increase apoptosis of H3255 NSCLC cells. H3255 cells were seeded on coverslips and treated for 48 hours with different EGFR-specific TKIs (erlotinib, 40 nM; osimertinib, 40 nM, or afatinib, 10 nM), either alone or in combination with 2XmAbs (cetuximab and trastuzumab, 5 µg/ml each). Cells were fixed in paraformaldehyde (4%) and incubated with an anti-cleaved caspase 3 antibody (CC3), followed by an Alexa Fluor 488-conjugated secondary antibody. DAPI (blue) was used to stain nuclei. Images were captured using a confocal microscope (20X magnification). Bar, 20 µm. Significance was assessed using one-way ANOVA followed by Tukey's multiple comparisons test. Note that non-significant comparisons are not indicated. All p-values corresponding to the data are listed in Appendix Table S2.

5. "Notably, due to relatively high abundance of BIM and cleaved caspase 3 in the control mice, we were unable to detect alterations in these markers of apoptosis." For the in vivo analyses, it would have been highly informative to analyze proliferation status of tumor cells.

High levels of apoptosis mediated cell turnover within tumors, even in the absence of treatment, is not surprising - but it makes it more challenging to interpret the data. Since growth and regression of tumors reflect net balance between proliferation and death, it would have been highly useful to complement the analyses shown in Figure 4 by assessing proliferation levels within tumors. Ideally, proliferation and apoptosis data should be assessed by histological analyses (IF or IHC) - if authors have samples of fixed tumor tissue from these experiments. I think that the interpretation of the in vivo data after 7 days of treatment is more complicated than a short-term in vitro assay - as one deals with cells that avoided initial elimination, and cell cycle arrest induced by TKIs might reduce the proportion of cells where apoptosis can be induced by the antibodies - thus even decrease in % of apoptotic cells in the combination group compared to the Ab one might be still compatible with enhanced cell death rates in the combination treatment.

We agree; analyzing regressing tumors in animals is highly challenging due to the dynamic changes and inter-animal variation.

As requested by this referee, we performed in tumors immunofluorescence analyses that used both a marker of apoptosis, cleaved caspase 3 (Figure XI; Appendix Figures S4A and S4B of the revised manuscript), and a marker of proliferation, Ki67 (Figure XII; Appendix Figures S4C and S4D of the revised manuscript). The tumors we used were derived from the mice shown in Figures 4 and EV6 of the original manuscript (Figures 4 and EV4 of the revised manuscript). Unfortunately, due to the rapid changes induced by the week-long treatments, some of the tumor tissues presented in Figure 4 were lost.

Figure XI (Appendix Figures S4A and S4B of the revised manuscript): Combinations of antibodies and TKIs increase apoptosis in vivo. CD1 nu/nu mice bearing PC9 xenografts were treated for one week with imatinib (100

mg/kg/day), erlotinib (50 mg/kg), either alone or in combination with 2XmAbs (cetuximab and trastuzumab, each at 0.1 mg/mouse/injection). Note that when singly applied, we used erlotinib at 50 mg/kg. Sections (4 μ m) were obtained from the following tumors: Control: mouse 3; 2XmAbs - mouse 1; erlotinib, mouse 2; erlotinib 50mg/kg+2XmAbs: mouse 2 (all from Figure 4), and both imatinib (mouse 3) and imatinib+2XmAbs (mouse 1; both from Figure EV4). Tumor sections were analyzed by means of immunofluorescence analysis, which utilized a rabbit anti-cleaved caspase 3 (CC3) antibody and a Cy3-secondary antibody (pseudo colored in red). CC3-positive cells were counted using the Fiji software. DAPI was used to stain nuclei. Images were captured using a fluorescence microscope (24X magnification). Bar, 50 μ m.

Figure XII (Appendix Figures S4C and S4D of the revised manuscript): Analysis of a proliferation marker, Ki67, in tumors undergoing treatment. Mice were treated as in Figures 4 and EV6 of the original manuscript. Sections (4 μ m) were obtained from the following tumors: Control (mouse 3), 2XmAbs (mouse 1), erlotinib (mouse 2), erlotinib 20+2XmAbs (mouse 2), erlotinib 50 mg/kg+2XmAbs (mouse 2 in Figure 4), and Imatinib (mouse 3), imatinib+2XmAbs (mouse 1 in Figure EV6). Tumor sections were analyzed by means of immunofluorescence, which

utilized an anti-Ki67 antibody and a Cy3-secondary antibody. Ki67-positive cells were counted using the Fiji software. DAPI was used to stain nuclei. Images were captured using a confocal microscope (20X magnification). Bars, 20 μ m.

6. For the RNA seq data presented in Figure 7 and Supplement, insufficient technical detail is provided. What are the criteria for the differential expression? Interestingly, there is no obvious difference between Erlotinib responding and non-responding tumors, in expression of genes differentially expressed between the controls and the combination group. Why only downregulated genes are analyzed for the Pathway analysis? By itself, the expression data does not enable to make strong conclusions that the authors make in terms of the underlying mechanisms. While functional validation would be ideal, at the very least the authors can dig deeper into the expression data that they already have.

In response to this comment, we better analyzed the expression data, revised the text of the manuscript and changed Expanded View Figure EV5.

(i) We selected genes that were differentially expressed between the Control and the Erlotinib+2XmAbs group, with a Log Fold Change (LFC) of at least +/- 1 (log2) and an adjusted p-value smaller or equal to, 0.000005.

(ii) According to our analysis, 19 genes were differentially expressed between erlotinib-responding and non-responding tumors (under less stringent conditions: LFC of at least +/- 1 and adjusted p-value < 0.05). We observed that 14 out of the 19 genes were also differentially expressed between the Erlotinib+2XmAbs group and the Control (untreated) group using the more stringent parameters to define the differentially expressed genes. The reason for using parameters that are more stringent for the comparison between the Erlotinib+2XmAbs group and the Control group is that the Control mice were collected in an earlier timepoint and did not receive treatment. This group could have more technical noise compared to the Erlotinib Responding and Erlotinib Resistant groups that were treated and maintained for comparable times.

(iii) We included all of the differentially expressed genes (n=74) in the pathway enrichment analysis, whenever they were upregulated or downregulated (Figure XIII; EV5-B of the revised manuscript). In addition, we modified the corresponding text of the revised manuscript. Essentially, all genes that were differentially expressed (both up- and down-regulated) were analyzed, but only genes relevant to cell cycle progression were presented. These genes were downregulated by the treatment, especially by the combined treatment with erlotinib+2XmAbs. Below we present validation, using qPCR, of an additional group of genes (Fig. XIV). All 5 genes presented below were up-regulated following treatment with the combination of drugs.

Figure XIII (Figure EV5-B of the revised manuscript): Combining two antibodies and erlotinib overcomes resistance to the TKI by modulating gene expression. PC9 xenografts were established and treated as described in Figure 7B of the manuscript. RNA was extracted from all tumors and utilized for RNA-seq analysis. Differentially expressed (DE) genes in the erlotinib+2XmAbs arm, compared to the control group, are presented in the pathway enrichment analysis using the NCATS BioPlanet (over-representation). The x-axis shows the negative logarithm of the enrichment adjusted p-value.

Figure XIV: Validation of differential gene expression effects. RNA was extracted from the tumors shown in Figure 7A of the manuscript and analyzed using qPCR. Primers specific to the indicated transcripts were used. Averages \pm SD of 2-5 mice are shown. GAPDH was used for normalization. Shown are: control (orange), responding to erlotinib (green), resistant to erlotinib (blue) and responding to erlotinib+2XmAbs (purple).

Minor comments

1. "No previous studies examined emergence of resistance to mAbs, or mAbs+TKI combinations" - this is clearly wrong (at least as stated, perhaps authors had something else in mind - in which case a more careful wording is warranted). There is plenty of literature on mechanisms of resistance to trastuzumab and other antibody-based therapies.

As requested, we re-phrased this sentence in the revised manuscript.

2. Results, first paragraph, page 4, line 7-8, "EGFR-specific TKIs" is unnecessarily repeated in the sentence.

The repeated words "EGFR-specific TKIs" were removed from the corresponding text of the revised manuscript.

3. Authors use different concentrations of TKIs in different in vivo experiments (such as 5 and 10 mg/kg osi), without providing a rationale for these differences.

The referee rightly refers to a point we explained in the revised manuscript: whereas erlotinib was applied in our study at the commonly used dose (50 mg/kg), we variably applied osimertinib at either 5 or 10 mg/kg. This was due to the relatively high efficacy and low toxicity of osimertinib in our animal models. Our initial tests (Figures 2 and 3) used osimertinib at the low dose, 5 mg/kg. Because at this dose osimertinib+2XmAbs was inferior to erlotinib (50 mg/kg)+2XmAbs, but in clinical studies osimertinib monotherapy better inhibited tumors, we later decided to increase the dose of osimertinib to 10 mg/kg. As expected, when we used osimertinib at the higher dose (Figure EV3), there was no difference between the two combination groups (osimertinib+2XmAbs and erlotinib+2XmAbs). As requested, we briefly explained these considerations in the revised manuscript.

4. It would have been highly helpful to show IdU-PI plots to characterize the impact of different treatments on cell cycle progression (in EV3). This would be more informative than using IdU by itself.

We note that CyTOF delineates cell cycle stages utilizing 5-iodo-2-deoxyuridine (IdU) to mark cells in S phase, simultaneously with antibodies against cyclin B1, phosphorylated retinoblastoma protein (pRb), and phosphorylated histone H3 (Ser28), which characterize the other cell cycle phases. For example, an antibody against pRb (Ser807 and 811) was used to separate cells found in the G0 and G1 phases. Specifically, we plotted Cisplatin vs. Iridium and selected the population of live cells. The latter were analyzed in the following ways:

- (i) IdU vs. Cyclin B1, to detect cells in G0/G1, S phase and G2/M
- (ii) IdU vs. pRb to detect cells in G0, and
- (iii) IdU vs. phospho-histone H3 to detect cells in the M-phase

We assume that propidium iodide, which labels DNA, might be used, but this option is not offered by our CyTOF Unit. In response to this comment, we better explained the analysis and cited the corresponding reference (Behbehani et al., 2012) in the text of the revised manuscript.

5. Consider being consistent when referring to the Expanded view data, as both Expanded view of figures (EV) and supplementary figures (S) notations are used in the main text. e.g, Page 6, second paragraph line 7, Figure S4--EV4; Page 10, line3, S7 -EV7; Page12, line 9, S8A-EV8A;

We corrected all these points in the revised form of the manuscript; there are five EV figures and 7 Appendix Figures in the revised manuscript.

6. Consider using consistent notation of TKI+2XmAbs throughout the paper when the combination treatment was conducted.

In the revised manuscript we refer only to TKI+2XmAbs when dealing with the combination treatment.

7. EV1 legend refers to the non-existent panel E, instead of the C.

The corresponding legend has been modified in the revised manuscript.

8. Consider indicating the cell line or xenografts name on western blot images throughout the results figures sections as authors did on Fig1A.

We revised most figures of the manuscript according to this comment.

9. "all combos associated with downregulation of IGF1R" - "combo" is overly colloquial.

We refrained from using the word “combo” in the revised text.

Referee #3:

This is an impressive paper representing a very complete study of EGFR inhibitor resistance in mouse models. There are a lot of animal expts to unpack here I commend the authors on the rigor of the study. The major mechanistic insight is that the combination of TKI+2xMABS cause a more pronounced inhibition of pERK and to some extent pAKT in vivo, causing extended anti-tumor control. This effect is mediated by proposed blocking of feedback loops that cause re-activation of this pathway when using EGFR TKIs alone. This is a potentially suitable paper provided some more in depth analysis can be done to elucidate the mechanistic basis for the triple drug combo.

Note: As per the instructions we received, our responses below are focused on comments 1, 4 and 5.

1.Perhaps the most pressing issue raised by this manuscript is a need for a more complete molecular characterization of mechanisms of resistance to EGFR TKIs in vivo (Fig 5, 6C). Given the efficacy of the triple combo, it appears the basic mechanism of resistance assumed to be at play in these settings is reactivation of EGFR an/or HER2. This needs to be demonstrated by analysis of relapsed tumor tissue for these phospho-proteins and downstream pathways. Without this knowledge one wonders whether the effects observed are truly on-target

We agree that reactivation of EGFR and/or HER2 might underlie resistance in our animal models. Hence, it would have been of great interest to assess the phosphorylation state of EGFR and HER2 in the relapsing tumors (Figures 5 and 6C). Unfortunately, from the studies presented in Figures 5 and 6C we do not have the relapsing tumors, since all tumors were re-treated and they eventually regressed. Nevertheless, we performed another experiment in animals injected with PC9 cells (shown in Figure XV; below). As predicted by the referee, we observed partial reactivation of EGFR and HER2 in relapsing tumors. In addition, we observed upregulation of additional RTKs (i.e., AXL, MET and IGF1R), along with sustained activation of ERK and AKT, and increased levels of survivin, the anti-apoptosis marker.

Figure XV: Resistance to first-line treatments (T790 wild type) associates with up-regulation of survival receptors, as well as with activation of ERK and AKT (see Appendix Figure S3). Mice harboring PC9 xenografts were treated for 54 days with monoclonal antibodies (2XmAbs; cetuximab plus trastuzumab, 0.2 mg/mouse/injection) once every three days, either alone or in combination with EGFR-specific TKIs, erlotinib (50 mg/kg/day) or osimertinib

(5 mg/kg/day, or 1 mg/kg/day when delivered in combination with antibodies). (A) Resistant or relapsed tumors were analyzed using immunoblotting and the indicated antibodies. (B) Representation of tumor volumes corresponding to individual animals analyzed in A.

2. Fig 5 -In vivo sequential treatments suggest that resistance to TKIs is mediated by EGFR/HER2 and respond to 2xMabs. This is shown in the PC9 model where such activity may be explained by the emergence of t790M in the case of Erlotinib, or something similar in the case of Osimertinib (perhaps C797S). Can the authors do sequence analysis of the in vivo resistant tumors to identify potential drivers of resistance in this model?

Although we had no samples left from the tumors shown in Figure 5, we were still able to perform the requested analyses on the tumors presented in Figures 7 and EV5. Exons 19 and 20 of EGFR were sequenced in these tumors. However, although we could confirm the presence of exon 19 deletion, we were unable to detect any point mutation in exon 20, neither T790M nor C797S (see Table I, below). The revised text refers to our inability to detect secondary mutations in exon 20.

	Exon 19 del E746_A750	Exon 20 T790	Exon 20 C797
Sequence	...CAA GGAATTAAGAGAAGCAACATC ACGCAGGGCT GCCTC ...
Control mouse 1	...CAA-----AAC...	...ATC ACGCAGGGCT GCCTC ...
Control mouse 2	...CAA-----AAC...	...ATC ACGCAGGGCT GCCTC ...
Control mouse 3	...CAA-----AAC...	...ATC ACGCAGGGCT GCCTC ...
Control mouse 4	...CAA-----AAC...	...ATC ACGCAGGGCT GCCTC ...
Control mouse 5	...CAA-----AAC...	...ATC ACGCAGGGCT GCCTC ...
Er-responding mouse 1	...CAA-----AAC...	...ATC ACGCAGGGCT GCCTC ...
Er-responding mouse 2	...CAA-----AAC...	...ATC ACGCAGGGCT GCCTC ...
Er-responding mouse 3	...CAA-----AAC...	...ATC ACGCAGGGCT GCCTC ...
Er-resistant mouse 1	...CAA-----AAC...	...ATC ACGCAGGGCT GCCTC ...
Er-resistant mouse 2	...CAA-----AAC...	...ATC ACGCAGGGCT GCCTC ...
Er-resistant mouse 3	...CAA-----AAC...	...ATC ACGCAGGGCT GCCTC ...
Combo-responding mouse 1	...CAA-----AAC...	...ATC ACGCAGGGCT GCCTC ...
Combo-responding mouse 2	...CAA-----AAC...	...ATC ACGCAGGGCT GCCTC ...

Table I: No secondary exon 20 mutations are detectable in drug-treated tumors. Complementary cDNA was synthesized from RNA previously extracted from the tumors presented in Figures 7 and EV5 of the revised manuscript using the qScript cDNA Synthesis Kit (Quantabio). Exons 19 and 20 of EGFR were amplified using the iProof High-Fidelity PCR kit (Bio-Rad), while using the following primers:
 Exon 19 forward: 5'-GCAATATCAGCCTTAGGTGCGGCTC-3'
 Exon 19 reverse: 5'-CATAGAAAGTGAACATTTAGGATGTG-3'
 Exon 20 forward: 5'-CCATGAGTACGTATTTTGAAACTC-3'
 Exon 20 reverse: 5'-CATATCCCCATGGCAAACCTTGC-3'
 PCR products were purified using QIAquick PCR Purification Kit (QIAGEN) and sequenced using the 3730 DNA Analyzer (from ABI).

3. The in vivo sequential study is highly dependent on the nature of drug resistance mechanism that may be different between models. This is critical to understand the breadth of tumor control expected with this combination, one could test this by applying a similar approach to Fig 5 with another model

We agree; performing this kind of experiments in different models (e.g., a PDX model) would be highly informative. Unfortunately, such experiments would require very long time (>5 months).

4. Fig 6- analysis of relapsed tumors indicating that they remain dependent on EGFR/HER2 in panel C. These studies are not analogous to residual disease in patients or in cell lines (eg

Hata et al, Ramirez et al) as they are under continuous drug treatment which is not true in this case. Clearly the residual disease state is not dependent on EGFR/HER2 because the cells that repopulate the tumor survive the initial treatment with Er+2xmABS. Hence it seems that drug holiday after residual disease leads to re-emergence of dependence on EGFR/HER2, but this dependency is not present at the RD stage itself. Analysis of signaling through serial sampling (upfront, RD, relapse) can be used to determine when EGFR/HER2 are most active.

We agree that the relapsed tumors analyzed in Figure 6C are not analogous to the residual fraction of cells reported by Michael Ramirez (Ramirez et al., 2016) and Aaron Hata (Hata et al., 2016). We also agree that after initial treatment with the combination of erlotinib+2XmAbs, only a few cells survive (possibly cells that have entered a persister state), but after treatment was stopped the initial population might have taken over once again, thus explaining the sensitivity of the tumors when the same treatment was re-applied. However, because all tumors shown in Fig. 6C were re-treated, we do not have samples from each stage of the experiment. Hence, analysis of signaling through serial sampling would require performing a completely new animal and relatively long experiment.

5. Fig 7, what happens to HER3 in this model? Consistent with RTK reactivation due to loss of negative feedback RTKs should be upregulated - please confirm. Also, does the Er-resitant tumor show reactivation of EGFR and/or activation of HER2? This is critical for explaining the mechanism of 2XmAbs.

To answer these queries, we re-analyzed tumor extracts using western blotting. The results are presented below, in Figure XVI. The new data were added and discussed in the revised manuscript (Appendix Figure S6).

- (i) As predicted by the referee, HER3 underwent up-regulation in this model.
- (ii) Likewise, the new data indicate that EGFR and HER2 underwent partial activation in tumors that acquired resistance to erlotinib.

PC9 xenografts

Figure XVI: By downregulating EGFR, HER2 and additional RTKs, a combination of two antibodies and erlotinib overcomes drug resistance in an animal model. Protein extracts were prepared from the tumors presented in Figure 7. The extracts were analyzed using immunoblotting. Signals were quantified and normalized to the signals corresponding to GAPDH. Numerical signals are shown below each lane.

Appendix: Quantification of all western blots presented

New panels inserted in Figure 1A: NSCLC expressing single site mutants of EGFR, PC9 (3×10^6) or H3255 (8×10^6), were seeded in 10-cm dishes. On the next day, complete media were replaced with media containing serum (1%) and the cells were treated for 24 hours with different EGFR-specific TKIs (erlotinib, 50 nM; osimertinib, 50 nM, or afatinib, 10 nM), either alone or in combination with 2XmAbs (cetuximab and trastuzumab, 5 μ g/ml each). Thereafter, cells were washed with cold saline and extracted. Proteins were separated using gel electrophoresis and transferred onto nitrocellulose membranes. After blocking, membranes were incubated overnight with the indicated primary antibodies, followed by incubation with horseradish peroxidase-conjugated secondary antibodies (60 minutes), and treatment with Clarity™ Western ECL Blotting Substrates (Bio-Rad). ECL signals were detected using the ChemiDoc™ Imaging System (Bio-Rad) and images were acquired using the ImageLab software. Signals (relative to Control) were quantified and normalized to the signals of GAPDH (numbers shown below each lane).

New panels inserted in Figure 4B: After treatment (see Figure 4A of the original manuscript), all mice were sacrificed and tumors extracted. Protein extracts were resolved by means of electrophoresis and transfer to nitrocellulose membranes, which were later incubated overnight with the indicated antibodies. This was followed by incubation with peroxidase-conjugated secondary antibodies. Signals were detected using Chemidoc™ (from Bio-Rad), quantified and normalized to the signals of GAPDH or tubulin (numbers shown below each lane).

New panel inserted in Figure 7C: Protein extracts were prepared from all tumors (this refers to Figure 7B of the original manuscript) and analyzed using immunoblotting, as indicated. Signals were quantified and normalized to the signals of GAPDH (numbers shown below each lane).

New panels inserted in Expanded View Figure EV1-B: PC9 cells were treated for 48 hours with increasing concentrations of EGFR-specific TKIs (erlotinib or osimertinib at 10, 20, 40, 80 and 160 nM; afatinib at 1, 5, 10, 20 and 40 nM), either alone or in combination with 2XmAbs (cetuximab plus trastuzumab, each at 5 μ g/ml). Protein extracts were resolved, blotted and probed with antibodies specific to the indicated apoptosis markers. Tubulin (or GAPDH) was used as loading control. Signals (relative to Control) were quantified and normalized to the signals of GAPDH or Tubulin (numbers shown below each lane).

New panels inserted in Expanded View Figure 2B: H3255 and PC9 cells were treated for 60 minutes with 2XmAbs (cetuximab and trastuzumab, each at 5 μ g/ml), either alone or in combination with the following TKIs: erlotinib or osimertinib (each at 50 nM) or afatinib (10 nM). Protein extracts were immunoblotted using the indicated antibodies. Signals (relative to Control) were quantified and normalized to the signals of GAPDH (numbers shown below each lane).

New panels inserted in Expanded View Figure 2C: H3255 cells were treated for 48 hours with increasing concentrations of EGFR-specific TKIs (erlotinib or osimertinib at 5, 10, 20, 40 and 80 nM, or afatinib at 0.1, 0.5, 1, 2 and 10 nM), either alone or in combination with 2XmAbs (cetuximab and trastuzumab, each at 5 µg/ml). Protein extracts were resolved using electrophoresis, transferred to nitrocellulose membranes and blotted with the indicated antibodies. GAPDH was used as a loading control. Signals, relative to Control, were quantified and normalized to the signals of GAPDH (numbers shown below each lane).

PC9 xenografts

New panel inserted in Expanded View Figure EV4B: Immunoblots of extracts prepared from the animals and tumors presented in Figure EV4-A are shown. GAPDH and tubulin were used as measures of total protein loaded. Note that mice in the control and 2XmAbs-treated groups are presented in Figure 4. Signals were quantified and normalized to the signals of GAPDH or tubulin (numbers shown below each lane).

New panel inserted in Expanded View Figure EV4-C: PC9 cells were treated for 4 hours with increasing concentrations of different TKIs (erlotinib or osimertinib, at 12.5, 25 and 50 nM; afatinib, at 2.5, 5 and 10 nM; imatinib, at 12.5, 25 and 50 μM). Protein extracts were resolved and immunoblotted for the indicated proteins. Signals (relative to Control) were quantified and normalized to the signals of GAPDH (numbers shown below each lane).

New panel inserted in Expanded View Figure EV5-E: PC9 cells were treated for 24 or 48 hours as described in Figure EV5-D of the revised manuscript. Protein extracts were immunoblotted with the indicated antibodies. GAPDH was used to control the amount of loaded protein. Signals (relative to Control) were quantified and normalized to the signals of GAPDH (numbers shown below each lane).

Appendix Table S2: Statistical parameters corresponding to individual figures.

Figure	Groups	Symbol	p-Value	N1	N2
1B_EGFR	Control vs. 2XmAbs	**	0,0091	4	4
	Control vs. Erlotinib	ns	0,7136	4	4
	Control vs. Er+2XmAbs	**	0,0064	4	4
	Control vs. Osimertinib	ns	0,6805	4	4
	Control vs. Os+2XmAbs	**	0,002	4	4
	Control vs. Afatinib	ns	0,592	4	4

	Control vs. Af+2XmAbs	**	0,0031	4	4
1B_HER2	Control vs. 2XmAbs	ns	0,9979	4	4
	Control vs. Erlotinib	**	0,0015	4	4
	Control vs. Er+2XmAbs	ns	>0,9999	4	4
	Control vs. Osimertinib	ns	0,1342	4	4
	Control vs. Os+2XmAbs	ns	0,9999	4	4
	Control vs. Afatinib	*	0,0355	4	4
	Control vs. Af+2XmAbs	ns	0,9998	4	4
	1B_HER3	Control vs. 2XmAbs	****	<0,0001	4
Control vs. Erlotinib		****	<0,0001	4	4
Control vs. Er+2XmAbs		****	<0,0001	4	4
Control vs. Osimertinib		****	<0,0001	4	4
Control vs. Os+2XmAbs		****	<0,0001	4	4
Control vs. Afatinib		****	<0,0001	4	4
Control vs. Af+2XmAbs		****	<0,0001	4	4
2C	Control vs. 2XmAbs	ns	0,9541	7	5
	Control vs. Erlotinib 50 mg/kg	ns	>0,9999	7	8
	Control vs. Er 50 mg/kg+2XmAbs	****	<0,0001	7	9
	Control vs. Er 20mg/kg+2XmAbs	ns	0,9966	7	8
	Control vs. Er 10 mg/kg+2XmAbs	ns	0,8969	7	9
	Control vs. Er 5 mg/kg+2XmAbs	ns	0,2468	7	9
	Control vs. Er 1 mg/kg+2XmAbs	ns	0,9893	7	8
	2XmAbs vs. Erlotinib 50 mg/kg	ns	0,9842	5	8
	2XmAbs vs. Er 50 mg/kg+2XmAbs	**	0,002	5	9
	2XmAbs vs. Er 20mg/kg+2XmAbs	ns	0,9997	5	8
	2XmAbs vs. Er 10 mg/kg+2XmAbs	ns	>0,9999	5	9
	2XmAbs vs. Er 5 mg/kg+2XmAbs	ns	0,9655	5	9
	2XmAbs vs. Er 1 mg/kg+2XmAbs	ns	>0,9999	5	8
	Erlotinib 50 mg/kg vs. Er 50 mg/kg+2XmAbs	****	<0,0001	8	9
	Erlotinib 50 mg/kg vs. Er 20mg/kg+2XmAbs	ns	0,9997	8	8
	Erlotinib 50 mg/kg vs. Er 10 mg/kg+2XmAbs	ns	0,9582	8	9
	Erlotinib 50 mg/kg vs. Er 5 mg/kg+2XmAbs	ns	0,3336	8	9
	Erlotinib 50 mg/kg vs. Er 1 mg/kg+2XmAbs	ns	0,9984	8	8
	Er 50 mg/kg+2XmAbs vs. Er 20mg/kg+2XmAbs	****	<0,0001	9	8
	Er 50 mg/kg+2XmAbs vs. Er 10 mg/kg+2XmAbs	***	0,0002	9	9
	Er 50 mg/kg+2XmAbs vs. Er 5 mg/kg+2XmAbs	**	0,0082	9	9
	Er 50 mg/kg+2XmAbs vs. Er 1 mg/kg+2XmAbs	****	<0,0001	9	8
	Er 20mg/kg+2XmAbs vs. Er 10 mg/kg+2XmAbs	ns	0,9988	8	9
	Er 20mg/kg+2XmAbs vs. Er 5 mg/kg+2XmAbs	ns	0,6399	8	9
	Er 20mg/kg+2XmAbs vs. Er 1 mg/kg+2XmAbs	ns	>0,9999	8	8
	Er 10 mg/kg+2XmAbs vs. Er 5 mg/kg+2XmAbs	ns	0,9237	9	9
Er 10 mg/kg+2XmAbs vs. Er 1 mg/kg+2XmAbs	ns	0,9998	9	8	
Er 5 mg/kg+2XmAbs vs. Er 1 mg/kg+2XmAbs	ns	0,7368	9	8	
Control vs. 2XmAbs	ns	0,9593	7	5	
Control vs. Osimertinib	ns	0,9989	7	8	

3C

Control vs. Os+2XmAbs	***	0,0001	7	9
Control vs. Afatinib	ns	0,2511	7	7
Control vs. Af+2XmAbs	**	0,0014	7	8
Control vs. Imatinib	ns	>0,9999	7	8
Control vs. Imat+2XmAbs	ns	0,99	7	8
2XmAbs vs. Osimertinib	ns	0,9991	5	8
2XmAbs vs. Os+2XmAbs	*	0,0241	5	9
2XmAbs vs. Afatinib	ns	0,9447	5	7
2XmAbs vs. Af+2XmAbs	ns	0,0972	5	8
2XmAbs vs. Imatinib	ns	0,9289	5	8
2XmAbs vs. Imat+2XmAbs	ns	>0,9999	5	8
Osimertinib vs. Os+2XmAbs	***	0,0007	8	9
Osimertinib vs. Afatinib	ns	0,5475	8	7
Osimertinib vs. Af+2XmAbs	**	0,0057	8	8
Osimertinib vs. Imatinib	ns	0,996	8	8
Osimertinib vs. Imat+2XmAbs	ns	>0,9999	8	8
Os+2XmAbs vs. Afatinib	ns	0,2287	9	7
Os+2XmAbs vs. Af+2XmAbs	ns	0,9992	9	8
Os+2XmAbs vs. Imatinib	****	<0,0001	9	8
Os+2XmAbs vs. Imat+2XmAbs	**	0,0016	9	8
Afatinib vs. Af+2XmAbs	ns	0,5664	7	8
Afatinib vs. Imatinib	ns	0,1761	7	8
Afatinib vs. Imat+2XmAbs	ns	0,7075	7	8
Af+2XmAbs vs. Imatinib	***	0,0006	8	8
Af+2XmAbs vs. Imat+2XmAbs	*	0,0121	8	8
Imatinib vs. Imat+2XmAbs	ns	0,9761	8	8
Control vs. 2XmAbs	ns	0,4834	4	7
Control vs. Erlotinib	ns	0,9978	4	8
Control vs. Er+2XmAbs	**	0,0034	4	10
Control vs. Osimertinib	ns	0,3583	4	10
Control vs. Os+2XmAbs	**	0,0084	4	10
2XmAbs vs. Erlotinib	ns	0,5742	7	8
2XmAbs vs. Er+2XmAbs	ns	0,1478	7	10
2XmAbs vs. Osimertinib	ns	>0,9999	7	10
2XmAbs vs. Os+2XmAbs	ns	0,2976	7	10
Erlotinib vs. Er+2XmAbs	***	0,001	8	10
Erlotinib vs. Osimertinib	ns	0,3999	8	10
Erlotinib vs. Os+2XmAbs	**	0,0032	8	10
Er+2XmAbs vs. Osimertinib	ns	0,1166	10	10
Er+2XmAbs vs. Os+2XmAbs	ns	0,9983	10	10
Osimertinib vs. Os+2XmAbs	ns	0,2619	10	10
Control vs. 2XmAbs	ns	0,9926	3	3
Control vs. Erlotinib	ns	0,403	3	3
Control vs. Er+2XmAbs	*	0,0292	3	3

Appendix S1B

Control vs. Osimertinib	ns	0,5251	3	3
Control vs. Os+2XmAbs	ns	0,3031	3	3
Control vs. Afatinib	*	0,0359	3	3
Control vs. Af+2XmAbs	**	0,0014	3	3
2XmAbs vs. Erlotinib	ns	0,8328	3	3
2XmAbs vs. Er+2XmAbs	ns	0,1203	3	3
2XmAbs vs. Osimertinib	ns	0,9185	3	3
2XmAbs vs. Os+2XmAbs	ns	0,7255	3	3
2XmAbs vs. Afatinib	ns	0,1451	3	3
2XmAbs vs. Af+2XmAbs	**	0,0062	3	3
Erlotinib vs. Er+2XmAbs	ns	0,7702	3	3
Erlotinib vs. Osimertinib	ns	>0,9999	3	3
Erlotinib vs. Os+2XmAbs	ns	>0,9999	3	3
Erlotinib vs. Afatinib	ns	0,8252	3	3
Erlotinib vs. Af+2XmAbs	ns	0,0938	3	3
Er+2XmAbs vs. Osimertinib	ns	0,6458	3	3
Er+2XmAbs vs. Os+2XmAbs	ns	0,8686	3	3
Er+2XmAbs vs. Afatinib	ns	>0,9999	3	3
Er+2XmAbs vs. Af+2XmAbs	ns	0,762	3	3
Osimertinib vs. Os+2XmAbs	ns	0,9998	3	3
Osimertinib vs. Afatinib	ns	0,7095	3	3
Osimertinib vs. Af+2XmAbs	ns	0,063	3	3
Os+2XmAbs vs. Afatinib	ns	0,9093	3	3
Os+2XmAbs vs. Af+2XmAbs	ns	0,1349	3	3
Afatinib vs. Af+2XmAbs	ns	0,7011	3	3

Appendix S1D

Control vs. 2XmAbs	ns	0,9679	3	3
Control vs. Erlotinib	ns	>0,9999	3	3
Control vs. Er+2XmAbs	ns	0,5811	3	3
Control vs. Osimertinib	ns	0,1355	3	3
Control vs. Os+2XmAbs	**	0,0028	3	3
Control vs. Afatinib	ns	0,5206	3	3
Control vs. Af+2XmAbs	*	0,0267	3	3
2XmAbs vs. Erlotinib	ns	0,9847	3	3
2XmAbs vs. Er+2XmAbs	ns	0,9838	3	3
2XmAbs vs. Osimertinib	ns	0,5637	3	3
2XmAbs vs. Os+2XmAbs	*	0,0194	3	3
2XmAbs vs. Afatinib	ns	0,9705	3	3
2XmAbs vs. Af+2XmAbs	ns	0,1646	3	3
Erlotinib vs. Er+2XmAbs	ns	0,6566	3	3
Erlotinib vs. Osimertinib	ns	0,1679	3	3
Erlotinib vs. Os+2XmAbs	**	0,0036	3	3
Erlotinib vs. Afatinib	ns	0,5957	3	3
Erlotinib vs. Af+2XmAbs	*	0,0341	3	3
Er+2XmAbs vs. Osimertinib	ns	0,9627	3	3

	Er+2XmAbs vs. Os+2XmAbs	ns	0,101	3	3
	Er+2XmAbs vs. Afatinib	ns	>0,9999	3	3
	Er+2XmAbs vs. Af+2XmAbs	ns	0,562	3	3
	Osimertinib vs. Os+2XmAbs	ns	0,4828	3	3
	Osimertinib vs. Afatinib	ns	0,9787	3	3
	Osimertinib vs. Af+2XmAbs	ns	0,9835	3	3
	Os+2XmAbs vs. Afatinib	ns	0,121	3	3
	Os+2XmAbs vs. Af+2XmAbs	ns	0,9309	3	3
	Afatinib vs. Af+2XmAbs	ns	0,623	3	3
Appendix S2C	Control vs. 2XmAbs	ns	>0,9999	3	3
	Control vs. Erl	ns	0,9515	3	3
	Control vs. Erl+2XmAbs	ns	0,4922	3	3
	Control vs. Osim	ns	0,2998	3	3
	Control vs. Osim+2XmAbs	ns	0,329	3	3
	Control vs. Afat	ns	0,3266	3	3
	Control vs. Afat+2XmAbs	ns	0,3261	3	3
Appendix S2D	Control vs. 2XmAbs	**	0,0055	3	3
	Control vs. Erl	****	<0,0001	3	3
	Control vs. Erl+2XmAbs	****	<0,0001	3	3
	Control vs. Osim	****	<0,0001	3	3
	Control vs. Osim+2XmAbs	****	<0,0001	3	3
	Control vs. Afat	****	<0,0001	3	3
	Control vs. Afat+2XmAbs	****	<0,0001	3	3
Appendix S2E	Control vs. 2XmAbs	ns	0,9999	3	3
	Control vs. Erl	ns	0,71	3	3
	Control vs. Erl+2XmAbs	ns	0,2913	3	3
	Control vs. Osim	ns	0,083	3	3
	Control vs. Osim+2XmAbs	ns	0,1711	3	3
	Control vs. Afat	ns	0,1102	3	3
	Control vs. Afat+2XmAbs	ns	0,1165	3	3
Appendix S3C	Control vs. 2XmAbs	ns	0,9953	5	7
	Control vs. Erlotinib	ns	>0,9999	5	7
	Control vs. Er+2XmAbs	**	0,0081	5	9
	Control vs. Osimertinib	ns	>0,9999	5	7
	Control vs. Os+2XmAbs	ns	0,9985	5	9
	Control vs. Afatinib	ns	>0,9999	5	8
	Control vs. Af+2XmAbs	ns	0,9035	5	9
	2XmAbs vs. Erlotinib	ns	0,9978	7	7
	2XmAbs vs. Er+2XmAbs	*	0,0293	7	9
	2XmAbs vs. Osimertinib	ns	0,9988	7	7
	2XmAbs vs. Os+2XmAbs	ns	>0,9999	7	9
	2XmAbs vs. Afatinib	ns	0,9734	7	8
	2XmAbs vs. Af+2XmAbs	ns	0,9992	7	9
Erlotinib vs. Er+2XmAbs	**	0,0039	7	9	

Erlotinib vs. Osimertinib	ns	>0,9999	7	7
Erlotinib vs. Os+2XmAbs	ns	0,9995	7	9
Erlotinib vs. Afatinib	ns	>0,9999	7	8
Erlotinib vs. Af+2XmAbs	ns	0,9126	7	9
Er+2XmAbs vs. Osimertinib	**	0,0047	9	7
Er+2XmAbs vs. Os+2XmAbs	**	0,009	9	9
Er+2XmAbs vs. Afatinib	***	0,0008	9	8
Er+2XmAbs vs. Af+2XmAbs	ns	0,0739	9	9
Osimertinib vs. Os+2XmAbs	ns	0,9998	7	9
Osimertinib vs. Afatinib	ns	>0,9999	7	8
Osimertinib vs. Af+2XmAbs	ns	0,9328	7	9
Os+2XmAbs vs. Afatinib	ns	0,9869	9	8
Os+2XmAbs vs. Af+2XmAbs	ns	0,9935	9	9
Afatinib vs. Af+2XmAbs	ns	0,7282	8	9

References

- Balana, M.E., Labriola, L., Salatino, M., Movsichoff, F., Peters, G., Charreau, E.H., and Elizalde, P.V. (2001). Activation of ErbB-2 via a hierarchical interaction between ErbB-2 and type I insulin-like growth factor receptor in mammary tumor cells. *Oncogene* 20, 34-47.
- Behbehani, G.K., Bendall, S.C., Clutter, M.R., Fantl, W.J., and Nolan, G.P. (2012). Single-cell mass cytometry adapted to measurements of the cell cycle. *Cytometry Part A : the journal of the International Society for Analytical Cytology* 81, 552-566.
- Brand, T.M., Iida, M., Corrigan, K.L., Braverman, C.M., Coan, J.P., Flanigan, B.G., Stein, A.P., Salgia, R., Rolff, J., Kimple, R.J., *et al.* (2017). The receptor tyrosine kinase AXL mediates nuclear translocation of the epidermal growth factor receptor. *Science signaling* 10.
- Hata, A.N., Niederst, M.J., Archibald, H.L., Gomez-Caraballo, M., Siddiqui, F.M., Mulvey, H.E., Maruvka, Y.E., Ji, F., Bhang, H.-e.C., Krishnamurthy Radhakrishna, V., *et al.* (2016). Tumor cells can follow distinct evolutionary paths to become resistant to epidermal growth factor receptor inhibition. *Nature medicine* 22, 262-269.
- Jo, M., Stolz, D.B., Esplen, J.E., Dorko, K., Michalopoulos, G.K., and Strom, S.C. (2000). Cross-talk between epidermal growth factor receptor and c-Met signal pathways in transformed cells. *J Biol Chem* 275, 8806-8811.
- Ramirez, M., Rajaram, S., Steininger, R.J., Osipchuk, D., Roth, M.A., Morinishi, L.S., Evans, L., Ji, W., Hsu, C.-H., Thurley, K., *et al.* (2016). Diverse drug-resistance mechanisms can emerge from drug-tolerant cancer persister cells. *Nature communications* 7, 10690.
- Sergina, N.V., Rausch, M., Wang, D., Blair, J., Hann, B., Shokat, K.M., and Moasser, M.M. (2007). Escape from HER-family tyrosine kinase inhibitor therapy by the kinase-inactive HER3. *Nature* 445, 437-441.

11th Jan 2021

Dear Prof. Yarden,

Thank you for the submission of your revised manuscript to EMBO Molecular Medicine, and please accept my apologies for the delay in getting back to you, which is due to the holiday season. We have now received the enclosed reports from the three referees who had reviewed your original manuscript. As you will see, while referees #1 and #2 are satisfied with the revisions, referee #3 still raises some minor concerns (please see the reports below):

Referee #3 comments:

- #3: While we agree with this referee that an additional model would strengthen the results, we also understand that this would take considerable time and effort. Therefore, if you do have data at hands, we would be happy for you to include it, but it will not be required for acceptance of the manuscript. In that case, please address this comment in writing and discuss potential limitations.
- #4: Please address this point along the lines indicated by the referee.

Furthermore, please address the following minor editorial issues:

1) Main manuscript text:

- Please answer/correct the changes suggested by our data editors in the main manuscript file (in track changes mode). This file will be sent to you in the next few days. Please use this file for any further modification.
- Please remove the red text.
- Material and methods:
 - o Antibodies: please indicate the dilutions used in the experiments.
 - o Cells: please indicate the origin and source of all cells used, and whether they were tested for mycoplasma contamination.
 - o Mice: please indicate the source, age and gender of the mice, as well as the housing and husbandry conditions.
- Please remove the "Materials and correspondence" section. As per our guidelines, it is understood that by publishing a paper in this journal, the authors agree to make available to colleagues in academic research all NEW reagents that were used in the research reported and that are not available from public repositories or commercial suppliers. As the H3255 cells were obtained from NCI and were published previously, the obligation is then on the primary paper to ensure distribution (please make sure to include citation of adequate literature).

2) Figures:

- Figure callouts: Please make sure that all figures and figure panels are referenced in the main text (panels from Fig. 2, Fig. 3 and Fig. EV3 are not called out).
- Please make sure that the text is readable, and that the resolution of the images is sufficient in your figures containing Western blots.
- Please add and/or define scale bars for all your panels containing immunofluorescence pictures, including in the Appendix figures.

3) Checklist: Please indicate the URL link and reference for your deposited dataset in section F/18.

4) As part of the EMBO Publications transparent editorial process initiative (see our Editorial at <http://embomolmed.embopress.org/content/2/9/329>), EMBO Molecular Medicine will publish online a Review Process File (RPF) to accompany accepted manuscripts.

In the event of acceptance, this file will be published in conjunction with your paper and will include the anonymous referee reports, your point-by-point response and all pertinent correspondence relating to the manuscript. Let us know whether you agree with the publication of the RPF and as here, IF YOU WANT TO REMOVE OR NOT ANY FIGURES from it prior to publication.

I look forward to receiving your revised manuscript.

Yours sincerely,

Lise Roth

Lise Roth, PhD
Editor
EMBO Molecular Medicine

To submit your manuscript, please follow this link:

Link Not Available

Photos 400-800 DPI

*Additional important information regarding figures and illustrations can be found at <https://bit.ly/EMBOPressFigurePreparationGuideline>

The system will prompt you to fill in your funding and payment information. This will allow Wiley to send you a quote for the article processing charge (APC) in case of acceptance. This quote takes into account any reduction or fee waivers that you may be eligible for. Authors do not need to pay any fees before their manuscript is accepted and transferred to our publisher.

***** Reviewer's comments *****

Referee #1 (Remarks for Author):

I had the opportunity to review the manuscript. I want to thank the authors in taking into account the previous reviews. It is clear they have made strong attempts to add the requested additional data. I am very happy with the changes and it has greatly strengthened the manuscript.

Referee #2 (Comments on Novelty/Model System for Author):

The mechanistic part of the manuscript is weak, but the animal system (main strengths of this paper) is adequate.

Referee #2 (Remarks for Author):

The authors have adequately addressed the comments. I think that the revised paper is suitable for publication without additional changes.

Referee #3 (Remarks for Author):

This is a strong manuscript and the rebuttal is adequate except for two of the points to be addressed:

3. While I appreciate that this will take time I believe that this is critical for the validity of the approach to overcome resistance. While I appreciate the use of PDX models to look at the upfront combination, only PC9 cells are used to examine the impact of the combination on overcoming acquired resistance. Addition a single additional model will demonstrate the breadth of effect and is important to assess the clinical validity of the conclusions.

4. In this case for the section "Residual disease remaining after treatment of a PDX model with mAbs+TKIs displays sensitivity to re-application of the drug combination" Please remove the term minimal residual disease throughout this section, as clinically this does not occur in the presence of a drug holiday, as the authors have modeled in Fig. 6C.

EMBO Molecular Medicine Manuscript: EMM-2020-13144

Ilaria Marrocco et al., *Upfront admixing antibodies and EGFR inhibitors preempts sequential treatments in lung cancer models*

Dear Dr. Lise Roth

Please find below our responses to the comments of the three Referees, especially the comments of Referee #3.

Referee #1 (Remarks for Author):

I had the opportunity to review the manuscript. I want to thank the authors in taking into account the previous reviews. It is clear they have made strong attempts to add the requested additional data. I am very happy with the changes and it has greatly strengthened the manuscript.

No additional changes have been requested.

Referee #2 (Comments on Novelty/Model System for Author):

The mechanistic part of the manuscript is weak, but the animal system (main strengths of this paper) is adequate.

Referee #2 (Remarks for Author):

The authors have adequately addressed the comments. I think that the revised paper is suitable for publication without additional changes.

No additional changes have been requested.

Referee #3 (Remarks for Author):

This is a strong manuscript and the rebuttal is adequate except for two of the points to be addressed:

3. While I appreciate that this will take time I believe that this is critical for the validity of the approach to overcome resistance. While I appreciate the use of PDX models to look at the upfront combination, only PC9 cells are used to examine the impact of the combination on overcoming acquired resistance. Addition a single additional model will demonstrate the breadth of effect and is important to assess the clinical validity of the conclusions.

We agree with the reviewer that repeating the experiment presented in Figure 5 of the manuscript using an additional cellular model would be relevant and helpful. Unfortunately, performing this kind of experiment would require many months: we will have to first make the tumors resistant to an EGFR-TKI and then apply the combination treatment. While we do not have such data on other models of first-line therapy, we performed and already reported a similar experiment performed with PC9ER cells (EGFR-mutated NSCLC cells resistant to erlotinib). In this experiment, nude mice bearing PC9ER xenografts were treated with osimertinib until tumors became resistant. Later, cetuximab and trastuzumab were added to the therapy, causing disease regression (see Figure 6 in Romaniello et al, [1]), as expected on the basis of our later studies using PC9 tumors.

4. In this case for the section "Residual disease remaining after treatment of a PDX model with

mAbs+TKIs displays sensitivity to re-application of the drug combination" Please remove the term minimal residual disease throughout this section, as clinically this does not occur in the presence of a drug holiday, as the authors have modeled in Fig. 6C.

We changed the text accordingly.

Reference

1. Romaniello, D., et al., *A Combination of Approved Antibodies Overcomes Resistance of Lung Cancer to Osimertinib by Blocking Bypass Pathways*. Clin Cancer Res, 2018. **24**(22): p. 5610-5621.

The authors performed the requested editorial changes.

2nd Feb 2021

Dear Prof. Yarden,

Thank you for sending the revised files. I looked at everything and all is fine. I am thus very pleased to accept your manuscript for publication in EMBO Molecular Medicine!

It will be sent to our publisher to be included in the next available issue of EMBO Molecular Medicine.

Please read below for additional important information regarding your article, its publication and the production process.

Congratulations on a nice study!

Yours sincerely,

Lise Roth

Lise Roth, Ph.D
Editor
EMBO Molecular Medicine

Corresponding Author Name: Yosef Yarden

Manuscript Number: EMM-2020-13144